

# A survey on graph neural networks, machine learning and deep learning techniques for time series applications in industry

Muhammad Jamal Ahmed, Alberto Mozo and Amit Karamchandani

E.T.S. de Ingeniería de Sistemas Informáticos, Universidad Politécnica de Madrid, Madrid, Spain

## ABSTRACT

Extensive studies have been conducted to investigate Artificial Intelligence (AI) in the context of time series data. In this article, we investigate the complex domain of industrial time series, from the dimensions of classical machine learning (ML), deep neural networks (DNNs) and graph neural networks (GNNs). Current surveys often focus on a specific methodology or oversee the connection of diverse approaches; our article bridges this gap by providing an all-inclusive interpretation across numerous techniques. In addition, the aim of this article is to focus on the core areas of time series such as forecasting, classification, and anomaly detection. From traditional methodologies like Autoregressive Integrated Moving Average (ARIMA) and support vector machine (SVM) methods, the advancements of DNNs, for instance long-short-term memory (LSTMs), convolutional neural networks (CNNs), attention mechanisms, and transformers, describe how temporal information is used for forecasting, anomaly detection, and classification. Then the article discusses the advances and limitations in ML, DNN, and GNN in order to improve the different methods in either category. Lastly, we outline future directions and open research questions with the different methodologies used in time series.

## INTRODUCTION

The advent of industry and the rapid expansion of today's industry 4.0 contribute to the increased complexity and automation of industrial processes. Although industrial processes on a large scale bring significant economic benefits to the country, the high level of interconnectivity and complexity within the structure means that even a minor disruption can result in serious fiscal setbacks and possibly loss of life. Therefore, to guarantee safe and reliable implementation of the manufacturing idea, fault prediction, detection, and diagnostic technology are crucial for all planned critical industrial productions. In *Palit & Popovic (2006)*, we explore the evolving network of use cases like autonomous robots, system integration, the Internet of Things, simulations, additive manufacturing, cloud computing, augmented reality, big data, and cyber security that outline the industrial quarter. As our environment becomes increasingly interconnected, the synthesis of digital know-how with conventional industrial methods has led to a

Corresponding author
Muhammad Jamal Ahmed,
muhammadjamal.a@upm.es

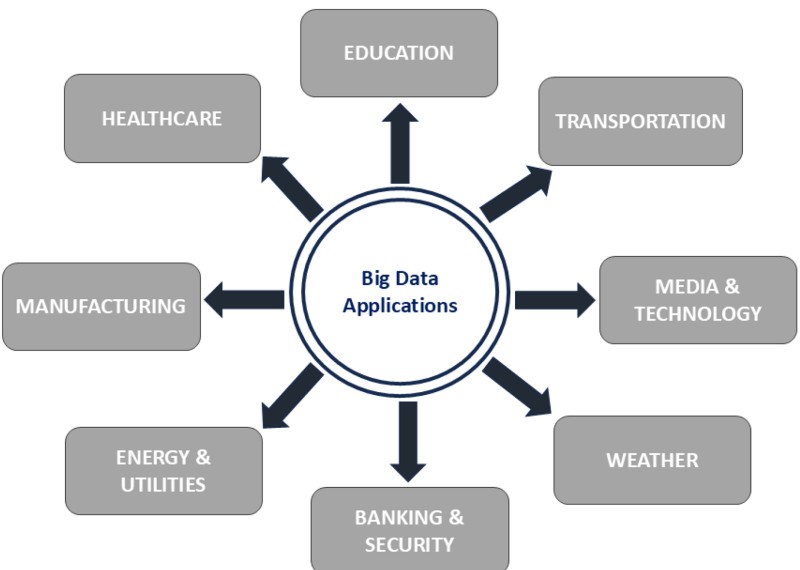

**Figure 1 Applications of big data in industrial sector.**

standard shift. Figure 1 condenses the diverse range of applications within Industry 4.0, where data analytics, smart systems, and automation converge to transform manufacturing technology and beyond.

The central driving force behind this advanced technological progress is the transformative power of big data. Figure 1 serves as a comprehensive guide to the numerous applications of big data within the industry 4.0, showcasing how data-driven insights are revolutionizing the global manufacturing sector. The integration of big data extends from predicting maintenance and optimizing supply chains to providing instant analytics and facilitating intelligent decision making. This incorporation of big data is pushing the boundaries of what is achievable in the industrial sector.

In the context of industrial environments, temporal analysis has become crucial for assessing operational reliability and efficiency and ensuring the safety and longevity of critical infrastructure. With the influx of temporal data from operational monitoring processes, the industrial sector is challenged to develop and integrate analytical approaches to effectively leverage this information to improve decision making and optimize its operational processes (*Bertolini et al., 2021*). Figure 2 shows a multistage approach to time series data analysis, where data is generated at the sensor level, undergoes initial processing and analysis, and is then transferred to the cloud to improve its accessibility and scalability.

Figure 3 illustrates the core components of the time series, each playing its own role in unraveling the anonymity of temporal drifts. From the all-encompassing course of trends to the periodic patterns of periods, to the surging sequences of cyclical effects, to the impulsive irregularities that add to the complexity, this figure guides the important workings that form the backbone of time series.

Classical machine learning, deep learning, and graph neural networks are established as standards for interpreting diverse patterns deeply rooted in time series data in industrial

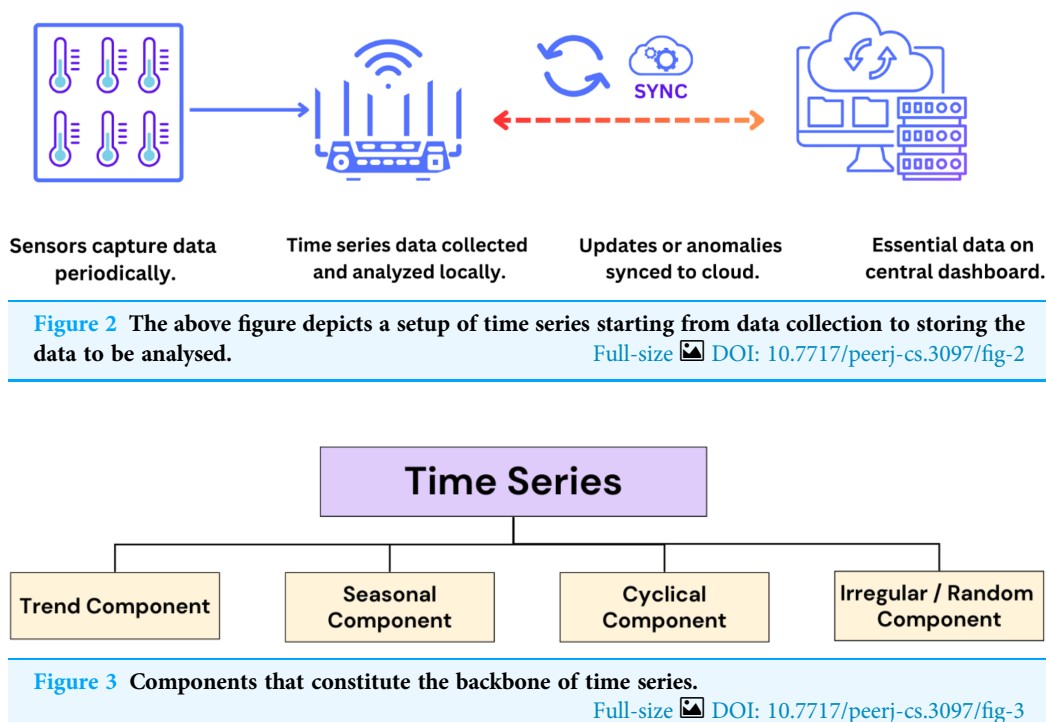

**Figure 2** The above figure depicts a setup of time series starting from data collection to storing the data to be analysed.

**Figure 3** Components that constitute the backbone of time series.

settings. In industrial settings, the origins of time series analysis can be traced back to the revolutionary efforts of *Box et al. (2015)*, who introduced the Autoregressive Integrated Moving Average (ARIMA) methods. These statistical techniques patented an elementary landmark in time series forecasting, providing the industrial sector with a robust structure to forecast demand, optimize production schedules, and maintain inventories. The advent of ARIMA and similar approaches represented a paradigm shift, allowing industry 4.0 to use historical data for informed policy making. Classical machine learning approaches, apparently signified by ARIMA, demonstrated an essential role in founding the early architectures for analyzing time series. These techniques, rooted in statistical fundamentals, propose real-world insights, and the interpretability that compiles them is especially valued for industries that require a vivid understanding of primary dynamics. In addition, traditional methods such as ARIMA have been shown to be effective in setups where data have strong temporal shapes and can be represented by means of nonlinear or linear associations (*Makridakis & Hibon, 2000*).

Recent advances in conventional machine learning practices have supported the incorporation of ensemble approaches for the analysis of time-series data. In particular, gradient-boosting machines and random forests have gained prominence. When dealing with nonlinear relationships and complex dependencies, ensemble models (*Karamchandani et al., 2023*) have seen progress within time series. By assembling predictions from different methods, these practices increase predictive capabilities and enhance robustness, making them convenient for various industrial frameworks (*Mozo et al., 2024*). Furthermore, ensemble models, conventional machine learning techniques, have progressed to integrate feature engineering methods, licensing improved illustration

of temporal dependencies and patterns. Feature engineering, coupled with typical processes such as support vector machine (SVM) and k-Nearest Neighbors (k-NN), has led to accurate improvements in predicting time-series tasks. These methods illustrate the adaptability of classical machine learning practices to face the shade limitations produced by practical industrial sector data in time series contexts. Although classical machine learning has made significant progress in the time series domain, it is important to acknowledge its limitations. The insights and interpretation in the real world offered by these techniques come at the cost of making inferences about the linearity and stationarity of the data (*Iglesias et al., 2023*). In situations where time series have complicated non-linear shapes or experience structural variations, typical machine learning approaches may fade.

Inscribing these boundaries requires a subtle method that fuses the power of conventional methods with evolving practices, certifying an adaptive and comprehensive framework for analyzing industrial time-series data. The field of time-series forecasting has undergone a groundbreaking metamorphosis with the arrival of deep learning techniques, with neural networks at the forefront. *Graves (2014)* presented an influential complement to this change with the construction of long short-term memories (LSTMs). These structures were designed to address the shortcomings of conventional techniques and to capture complicated temporal dependencies by allowing the time-series model to capture long-term dependencies in the data. The introduction of LSTM techniques has opened new avenues for dynamic and accurate prediction of time-series data, representing an innovation in the field of industrial environments. An essential milestone in the direction of deep learning for the prediction and analysis of time series data was manifested by the research of *Wang et al. (2018)*, who proposed sequence-to-sequence methods. This novelty was licensed to industries to predict failure detection and optimize complicated supply chain processes. Sequence-to-sequence methods had the power to channel input sequences and produce their output sequences, transforming predictive analytics and enlightening further understanding and comprehension of temporal dependencies and patterns. The metamorphic power of deep learning, as demonstrated by these developments, has made it an essential implement for addressing encounters modeled by practical time series data in industries (*Cutler et al., 2007*).

Current research by *Yang et al. (2021)* has explored attention architectures within deep learning methods for time series analysis. These mechanisms increase interpretability by allowing the technique to focus on explicit parts of the input sequence, thereby gaining an understanding of the aspects that drive the prediction. Approaches such as transfer learning, in which pre-trained methods are revised for precise industrial setups and adhere successfully to *Schölkopf et al. (2001)*. This method influences information from one domain to boost production in the other, an advantageous approach in situations where there is a limited amount of labeled data. Nevertheless, it is critical to admit the trials associated with deep learning, together with the need for considerable computational resources, the amount of labeled data, and possible overfitting in high-dimensional, multifaceted datasets.

Anomaly detection is a key player in maintaining the safety and reliability of industrial settings, certifying the early detection of faults before they intensify into serious problems. Traditional techniques, such as SVM and isolation forests of *Liu, Ting & Zhou (2012)*, have been the followers in this field, contributing active solutions for the recognition of eccentricities of usual behavior. However, the field has undergone a transformative change with the incorporation of deep learning practices, notably variational autoencoders (VAEs) (*Kingma & Welling, 2022*). These techniques have played and continue to play a critical role in improving anomaly detection capabilities in time series data for industrial frameworks. VAEs could learn the probabilistic distribution of standard data and, therefore, it can successfully classify subtle deviations that might indicate anomalous behavior. The current stream of anomaly detection studies has validated the use of generative adversarial networks (GANs) in this area, as verified by *Akcay, Atapour-Abarghouei & Breckon (2019)*, embodies the ability to integrate different deep learning practices to achieve more robust anomaly detection models. Furthermore, the integration of deep learning and attention mechanisms has shown potential in anomaly detection. A recent study by *Vaswani et al. (2017)* discovers the incorporation of attention approaches into time series, highlighting their role in illuminated anomaly detection performance in industrial settings.

*Li et al. (2018)* comes up with significant arguments by revealing the success of graph neural networks (GNNs) in the domain of time-series forecasting, exclusively in situations where time series data in an industrial context have characteristic graphical assemblies. GNNs go beyond traditional machine learning (ML) and thus captures these associations, which provides a more comprehensive understanding of the hidden dynamics particularly valuable in industries. A study by *Zhang et al. (2022a)* explores the integration of reinforcement learning with GNN, offering a pioneering exploration of their collective potential. Reinforcement learning, coupled with the ability of GNNs to handle complicated dependencies, generates a coactive method that embraces potential for augmenting decision approaches in response to the dynamic environments of industrial sectors. GNNs can successfully capture anomalous patterns that are evident in interconnected structures, giving a robust result for detecting deviations from normal behavior. The research in *Zhang et al. (2022a)* illustrates the positive incorporation of GNNs in the detection of anomalies, demonstrating the applicability of GNNs in several domains, taking into account the industrial context. Although GNNs offer exceptional capabilities, challenges remain in terms of interpretability and scalability. Addressing these challenges is crucial to certifying the unified incorporation of GNNs into the frameworks of industrial sectors (*De Livera, Hyndman & Snyder, 2011*). However, with the change in criteria towards neural networks, interpretability has emerged as a major concern in the industrial field. *De Livera, Hyndman & Snyder (2011)* highlighted the importance of interpretability in these networks and identified its implications for trusting and understanding the conclusions drawn by these approaches in crucial industrial situations.

The overview of attentional structures by *Vaswani et al. (2017)* proved to be an essential development, allowing practices to focus on serious time sections within time series. This mechanism not only improved overall accuracy, but also demonstrated good

interpretability for classification practices, confirming that important information is not ignored. The evolution of classification practices, as observed in the studies of *Taheri, Gimpel & Berger-Wolf (2019)*, indicates a constant determination to make even these methods compatible with the subtle requirements of classification of industrial activity. In pursuit of filtering classification duties for time series data in industrial frameworks, current exploration by *Li, Wu & Xu (2022)* discovers the life-changing potential success of transformer-based structures. The use of these architectures signifies a retreat from classical recurrent neural networks and presents a new viewpoint on the classification of different sequences in the industrial background. The exploration of transformer-based techniques highlights the adaptability and compliance of neural networks and their volume to evolve in aggregation with the exclusive challenges modeled by time series data in Industry 4.0. As industries more and more hold the incorporation of different practices, from traditional SVMs to front-line transformer techniques, the survey will steer precisely through apiece standard. The key aspects of our survey are as follows.

- Interdisciplinary synthesis: By linking conventional machine learning approaches, cutting-edge deep learning frameworks, and evolving graph neural network practices, this review forms an interdisciplinary description. This combination enables the reader to grasp the evolution of analyzing time-series data across different technological domains, providing a broader understanding.
- Comprehensive exploration: This survey not only explores the theoretical underpinnings of each architype but also offers practical intuitions into their real-world applications. This all-encompassing exploration provides the reader with a nuanced perspective that enables informed decision-making about the variety of methodologies built on explicit application or industry requirements.
- Emerging trends and challenges The advantages and challenges associated with each standard are addressed; this survey enunciates to the testimony of potential research directions and emerging trends. This progressive technique is helpful for practitioners, decision makers, and researchers, looking for a stay to take the modern advances in time series analysis.
- Applicability across industries: The survey's emphasis on industrial data adds a layer of distinction, making it especially relevant to industries that are transforming to the digital side. By illuminating how each area can be useful in industrial environments, this study helps as an applied guide for experts navigating the convolutions of time series within various segments. By being expressive, the data can be applied in industrial frameworks.
- Guidance for future research: The fusion of classical machine learning, deep learning, and graph neural networks, coupled with a thorough study of challenges, circles the phase for future research activities. This study not only fuses the current knowledge but also addresses the branded limitations.

In Table 1, we compare our survey with other surveys. Survey one (*Bertolini et al., 2021*) has just investigated ML techniques, the second survey (*Drakaki et al., 2022*) thoroughly focuses on ML and DL methods, the third one (*Waikhom & Patgiri, 2023*) clearly focuses

**Table 1** Comparisons of different survey articles with ours.

|  | ML techniques | DL techniques | GNNs |
| --- | --- | --- | --- |
| *Bertolini et al. (2021)* | ✓ | ✗ | ✗ |
| *Drakaki et al. (2022)* | ✓ | ✓ | ✗ |
| *Waikhom & Patgiri (2023)* | ✗ | ✗ | ✓ |
| Our survey | ✓ | ✓ | ✓ |

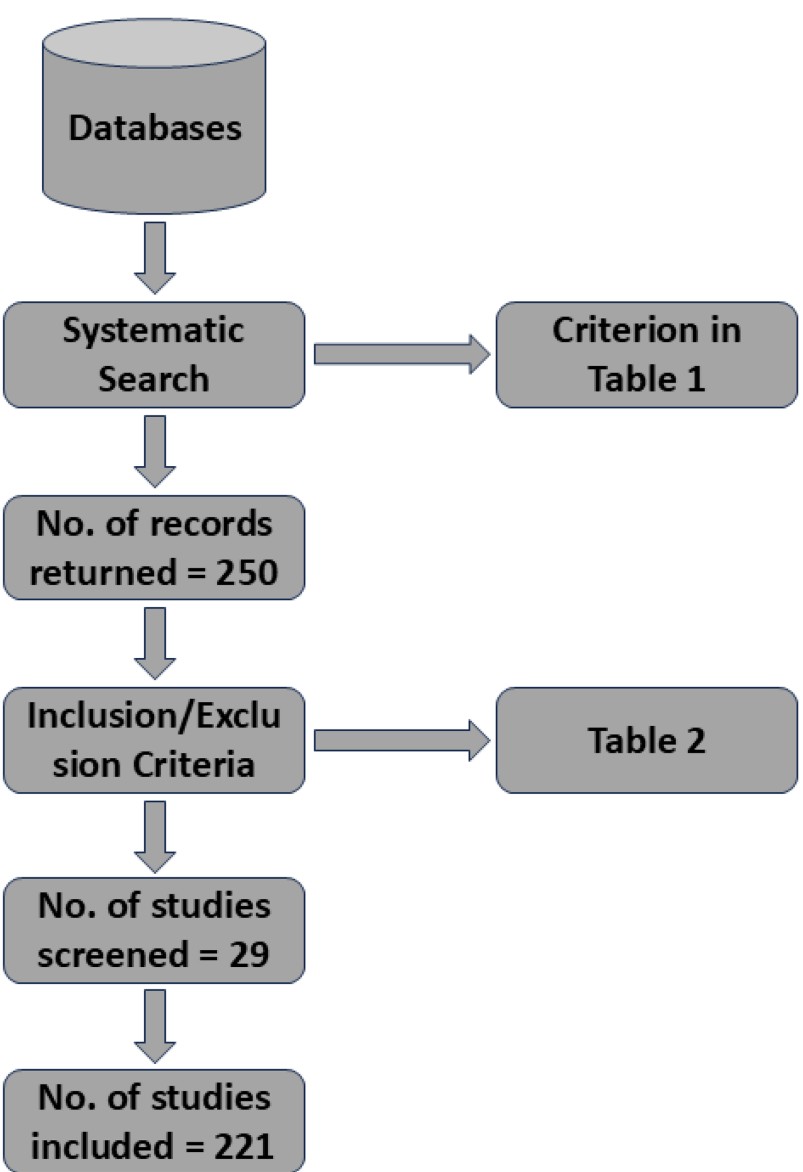

**Figure 4** PRISMA diagram to illustrate the study selection process.

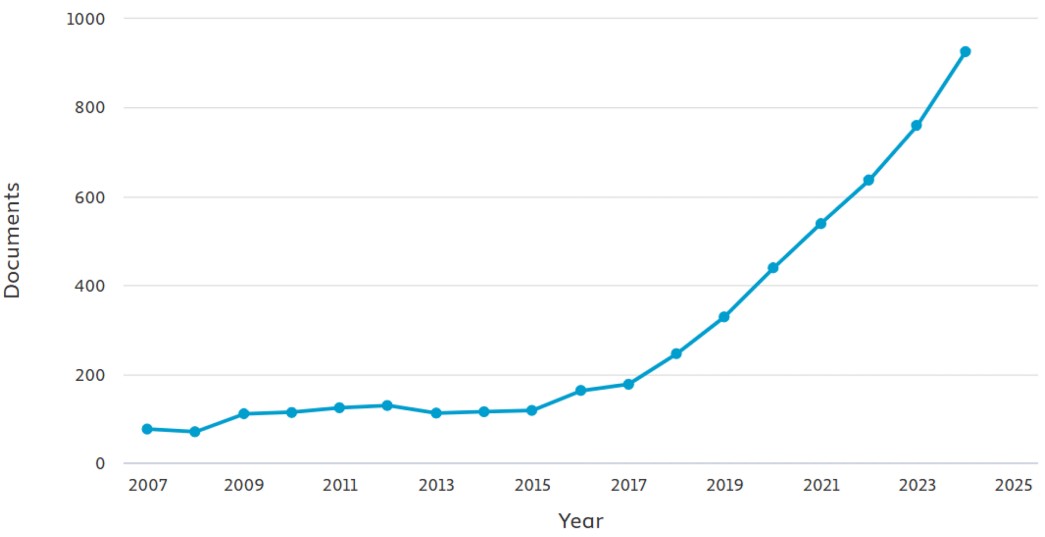

**Figure 5** **The number of documents *i.e.*, articles and conferences published yearly.** This search was performed on the Scopus platform.  

on GNN, compared to these surveys we have deep dived into all three techniques investigated in the surveys mentioned, which is a unique part of our survey.

## METHODOLOGY

This section provides a detailed description of the systematic approach used to conduct this study and the survey on time series analysis for industrial frameworks using graph neural networks, machine learning, and deep learning.

The literature initially collected was a total of 250 manuscripts. The literature was collected from different academic databases and sources, including Google Scholar, ArXiv, PubMed, ACM Digital Library, and IEEE Xplore. Various keywords such as ('time series analysis' OR 'forecast') AND ('machine learning' OR 'deep learning' OR 'graph neural networks') AND ('fault detection' OR 'anomaly detection'). Databases like Google Scholar are known for their broader coverage, ensuring that gray literature and interdisciplinary studies are not overlooked. The inclusion of ArXiv was to capture latest and cutting edge research, which may not be avaiable in peer-reviewed literature. The other databases such as IEEE Xplore and PubMed were given priority because of peer-reviewed and high-quality articles in engineering and computational fields. The selection of unsupervised and semi-supervised learning (SSL) models, was their emphasis on limited labeled data scenarios most common in industry. The exclusion consists of studies that lacked practical validation and focused solely theories in industry. In Fig. 4 we show the PRISMA diagram to illustrate the study selection process. In Fig. 5 we illustrates the yearly studies conducted and how these studies have rose in all these years. Tables 2 and 3 shows the systematic search of different articles and the inclusion as well as exclusion criteria of articles.

**Table 2  Systematic search on different academic databases (title, abstract, keywords).**

| Criterion | Search terms |
| --- | --- |
| ML/DL | ("Machine learning" OR "Deep learning" OR "Artificial Intelligence" OR "AI" OR "Neural Network") |
| TSC | (Classif* OR "Prediction" OR "Forecasting") AND ("Time series") |
| GNN | ("Graph neural network" OR "GNN" OR "Graph convolutional network" OR "GCN" OR "Graph attention network" OR "GAT") |
| Industry | ("Industrial Applications" OR "Manufacturing" OR "Smart Industry" OR "Industry 4.0") |
| Type | Journal article OR Conference article |
| Language | English |

**Table 3  Inclusion and exclusion criteria of our survey article (INCL stands for inclusion and EXCL stands for exclusion).**

| Criterion | Search terms |
| --- | --- |
| INCL 1 | Time series classification |
| INCL 2 | Time series Forecasting OR Prediction |
| INCL 3 | Time series Anomaly Detection |
| INCL 4 | Only articles that precisely address explainability or interpretability |
| INCL 5 | Only articles that show their approach for industrial time series |
| EXCL 1 | Articles those are published prior to 2007 |
| EXCL 2 | Articles without any citations |
| EXCL 3 | Articles that solely conduct analyses purely on the statistics |
| EXCL 4 | Articles those do not involve industry |

## APPLICATIONS OF TIME-SERIES ANALYSIS

### Time series forecasting

A critical feature of data exploration involves predicting or forecasting future values based on historical information. This method finds wide applications in various fields, including economics, finance, weather forecasting, energy utilization, and industrial manufacturing. The intention is to get an idea of the elemental trends and patterns within a timely methodical arrangement of data points to generate accurate predictions. In order to make effective predictions, we need to understand the different components of time series data, and these components are necessary to make compelling predictions. The three main components are: (a) Trends, the long-term direction or movement in the data. Trends can have no direction, can be down and can be up, (b) Seasonality—the fluctuations or frequent patterns in the time series data that occur at regular intervals, repeatedly, by aspects such as the time of the day, a few days, a week, a month, or a season, (c) Noise—the data that have arbitrary irregularities or fluctuations that are not described by the seasonality or the trend. It means the inherent unpredictability of pragmatic values.

### Time series anomaly detection

This effort focuses on identifying unexpected events and irregularities in an event within time-series or sequential data. Anomaly detection involves determining the timing of the abnormal event, while anomaly identification involves gaining significant insight into why,

where, and how the anomaly occurred. Because anomaly events are often difficult to obtain, recent research often approaches this task as an unsupervised task, which involves building a model that characterizes non-anomalous or normal data. This trained model is then used to identify anomalies by producing a good score each time an anomalous event is detected.

### Time series imputation

The focus is on the estimation and completion of missing or partial data points within a time series. Ongoing studies in this area can generally be divided into two primary methods: out-of-sample and in-sample imputation. Out-of-sample imputation deals with the imputation of missing data that are not available in the training data file, whereas in-sample imputation involves the imputation of missing values within a given time series data.

### Time series classification

The main goal of classification in time series data is to assign a categorical label to a given data by considering its characteristics or underlying patterns. Rather than capturing different sequences within individual time series samples, classification primarily involves recognizing discriminative patterns that help to separate samples based on their own class marker, the label.

## FOUNDATIONAL MODELS OF TIME SERIES ANALYSIS

Traditional machine learning models are often characterized by their shallowness. In the context of artificial neural networks (ANN), standard neural networks (NN) typically have no more than two layers and have limited data processing capabilities in their original state (*Choudhary et al., 2022*). In various operations, the need for prior feature engineering arises when examining extensive data, as well as data with high dimensionality, with traditional neural networks. Typically, data are processed using dimensionality reduction techniques, including data mapping techniques such as SOM or Principal Component Analysis (PCA) (*Gewers et al., 2021*). As a result, a hybrid intelligent system formed by combining two or more models becomes essential to effectively analyze complex data. Deep learning, as indicated by the use of many hidden layers within the architecture of an artificial neural network (ANN) (*Bui et al., 2020*), differs fundamentally from conventional machine learning models in the way in which it learns representations from raw data (*Angelov & Gu, 2019*). Several levels of abstraction (*Bengio, 2009*) are acquired for data representation in a deep learning model. Essentially, the learning process derives increased meaning from the data using advanced data abstraction at a higher level (*Yassine et al., 2019*). Deep learning methods defeat outdated machine learning techniques in the number of hidden layers. However, the distinctive and defining aspect of deep learning that distinguishes it from traditional machine learning is the advanced feature engineering capabilities of deep learning models. Table 4, presents the use cases, strengths, and limitations of different models in industrial sector. Table 5, shows the comparative analysis of different approaches of ML, DNN, and GNN in time series tasks. Table 6 shows a

**Table 4  Use cases, strengths, and limitations of different models in industrial sector.**

| Model type | References | Use Cases | Strengths | Weaknesses |
|---|---|---|---|---|
| ARIMA | *Xu et al. (2022)* | Time-series forecasting (*e.g.*, sales, energy demand) | Interpretable, effective for linear trends and seasonality | Limited to linear dependencies, struggles with non-stationary data |
| LSTM | *Xu et al. (2022)* | Predictive maintenance, anomaly detection in time-series | Captures long-term dependencies, handles sequential data | Computationally expensive, requires large datasets |
| CNN | *Wang et al. (2020a)* | Quality control using visual data, fault detection | Excels at spatial and hierarchical feature extraction | Struggles with sequential data, not inherently interpretable |
| GNN | *Chen, Feng & Wirjanto (2023)* | Supply chain optimization, industrial IoT, network fault diagnosis | Models relational and topological data, scales to complex networks | Computationally demanding, requires domain knowledge for graph construction |

**Table 5  Comparative analysis of ML, DNN, and GNN approaches in time series tasks.**

| Approach | Reference | Specific applications in time series | Computational complexity | Accuracy | Scalability | Strengths | Limitations |
|---|---|---|---|---|---|---|---|
| Machine learning (ML) | *Goldsteen et al. (2022)* | Forecasting: ARIMA, Classification: SVM, Anomaly Detection: Isolation Forest. | Relatively low for basic algorithms (*e.g.*, Linear Regression, SVM), but increases with ensemble methods (*e.g.*, Random Forest). | Moderate for simple models; varies with data preprocessing and feature engineering. | High, suitable for small to large datasets. | Easy to implement, interpretable results, low resource requirements. | Performance depends heavily on feature engineering; limited for highly complex datasets. |
| Deep neural networks (DNNs) | *Zintgraf et al. (2017)* | Forecasting: LSTMs, GRUs; Classification: CNNs, MLPs; Anomaly Detection: Autoencoders. | High, due to training and optimization of deep architectures. | High, especially for large and complex datasets. | Moderate, depending on hardware and parallelization. | Ability to model complex relationships, no need for feature engineering, effective for large-scale data. | Resource-intensive, requires significant labeled data and computational power; prone to overfitting. |
| Graph neural networks (GNNs) | *Zambon, Livi & Alippi (2022)* | Classification: Node or graph-level predictions; Anomaly Detection: Structural patterns in graphs. | High, involves graph construction and message-passing operations. | High, particularly for data with inherent graph structure or inter-dependencies. | Moderate, scalability depends on graph size and sparsity. | Captures spatial and relational data, handles irregular data structures, interprets complex dependencies. | Computationally expensive, challenging for dense graphs or datasets without clear graph structure. |

continuation of Table 5, depicts the different key performance metrics to better understand the strengths and limitations.

This uniqueness is characterized by intricate feature abstraction and construction, which is integrated into the assembly of the technique within the learning process. Robustness to data variation is a key strength of deep learning models, attributed to their

**Table 6 Analyzing key performance metrics to better understand strength and limitations.**

| Approach | Reference | Errors | Interpretability | Computational cost | Suitability for sequential data |
|---|---|---|---|---|---|
| ML | *Goldsteen et al. (2022)* | High RMSE | High | Low | Poor (non-stationary issues) |
| DNNs | *Zintgraf et al. (2017)* | Low MAE and RMSE, high F1-score | Moderate and Low | High | Excellent (long-term dependencies) |
| GNNs | *Zambon, Livi & Alippi (2022)* | Low MAE and RMSE, high F1-score | Moderate | High | Moderate (topological data) |

advanced intangible representation and feature engineering capabilities (*Wang et al., 2019*). In addition, the hierarchical structure inherent in deep networks enables them to effectively model the intricate nonlinear relationships present in large datasets. In contrast, traditional ML models often face challenges when dealing with large and high-dimensional data sets. As a result, ML models rely on feature selection as a dimensionality reduction method to facilitate the processing of large datasets more efficiently. The difficulty arises in real-world industrial applications because the large data sets collected are often contaminated with noise, outliers, and various types of anomaly, which poses a formidable challenge in feature selection. In the reign of intelligent engineering, autoencoders and their diversity, recurrent neural networks (RNNs), deep belief networks (DBNs), and convolutional neural networks (CNNs) stand out as the most widely used DL networks.

## Deep belief network

A deep belief network is formed by assembling several restricted Boltzmann machines (*Hinton, 2007*). It is worth noting that in a DBN there are connections between layers, whereas within a layer there are no connections between neurons. The construction of the network, organized layer by layer, facilitates the development of a classification feature representation (*Lagunas & Garces, 2017*). This representation is instrumental in constructing an advanced representation of the input information. The DBN achieves input reconstruction by learning a probability distribution in the unsupervised training process. The Restricted Boltzmann Machine is a propagative stochastic feedforward artificial neural network known for its effectiveness in feature engineering. Training a DBN involves training multiple RBMs. The lower RBM, the hidden layer, is considered the network's training data, and the output of that RBM serves as the training data for the higher RBM. Once all RBMs are trained, a fine-tuning procedure is performed using a backpropagation process with the training information as the output (*Lee et al., 2018*).

## RNNs for time series analysis

There are different classes of neural networks, one of which is RNNs illustrated in Fig. 6, which are designed to handle sequential and continuous data, making them particularly useful for analyzing time-series data. Unlike classical feedforward neural networks, RNNs have networks that formulate cycles, allowing them to maintain a hidden state that captures information about previous inputs into the system. RNNs take advantage of the hidden state to capture both temporal and contextual dependencies in sequential data. The

**Peer**J Computer Science

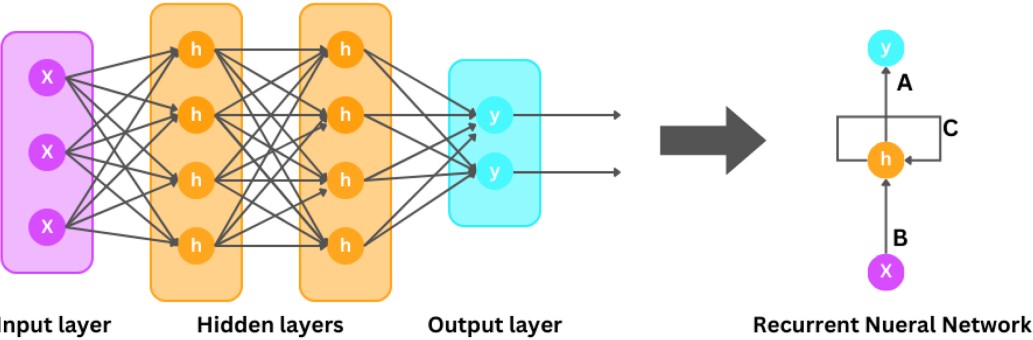

**Figure 6  Recurrent neural network.**

scientific design of an RNN involves computing the hidden states at each time interval based on the input information at that time step and also from the previous time step. Suppose that the input sequence is denoted as:

$$X = \{x^{(1)}, x^{(2)}, \ldots, x^{(T)}\}$$

where $x(t)$ characterizes the input at each time step t. Similarly, let $H = \{h^{(1)}, h^{(2)}, \ldots, h^{(T)}\}$ denote the hidden state sequence. The hidden state which is h(t) is calculated using the following equation.

$$h(t) = \sigma(W_{hx}x(t) + W_{hh}h(t-1) + b_h).$$

While $W_{hx}$ denotes the weight matrix for the input $x(t)$, $W_{hh}$ denotes the weight matrix for $h(t-1)$, which is again the hidden state, b h shows us the bias term, and $\sigma$ finally corresponds to the activation function, generally tanh or ReLU are the most commonly used activation functions. The hidden state is then used to generate the output at this time step:

$$y(t) = \sigma(W_{yh} \cdot h(t) + b_y).$$

While $W_{yh}$ represents the weight matrix for the connection between the output and the hidden state, $b_y$ represents the bias term. The RNN can be trained by adjusting the hyperparameters ($W_{hx}, W_{hh}, h_{bh}, h_{Wyh}, b_y$) to reduce the loss, which shows the amplitude of the variance between the actual target and the predicted output y (t).

However, traditional RNNs suffer from the limitations of exploding and vanishing gradients. The gradient vanishing trick happens when gradients become very small in the time of backpropagation, making it difficult for the model to learn long-range dependencies. Gradient exploding occurs when gradients are too large, causing unpredictability during training.

## Long short-term memory

To address the issues discussed in RNN, other advanced RNN designs, known as gated recurrent units (GRUs) and long short-term memory (LSTMs) illustrated in Fig. 7, have been brought into the light. LSTMs were designed to overcome the problems of traditional RNNs, namely the vanishing gradient dilemma, to better learn the long-term dependencies
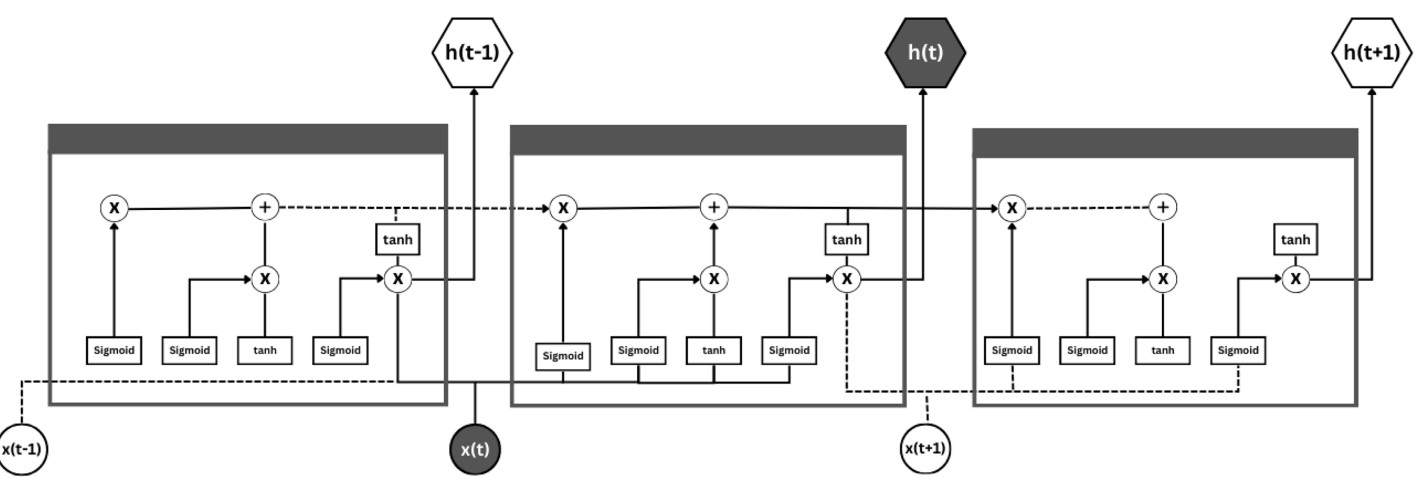

**Figure 7  Long short-term memory.**

in sequential time series data. In LSTMs, this is achieved by introducing gating mechanisms and memory cells to regulate the flow of information between networks.

The key idea is the memory cell, which collects and holds information for long periods of time. The main building blocks of the memory cell are structured by three gates: the input gate $(i_t)$, the forgetting gate $(f_t)$ and the output gate $(o_t)$. The purpose of these three gates is to manage the input and output of information. Suppose that at time step t, the input is $X_t$, at time step $t_1$, the hidden state is $H_{t-1}$, and the memory cell state is $C_{t-1}$ at time step t−1. Furthermore, the input value at time step t is assumed to be $X_t$, while at time step t−1 the hidden state value is $H_{t-1}$ and the memory cell state is $C_{t-1}$ at time step $\{t-1\}$.

The input gate $i_t = \sigma(W_i \cdot [H_{t-1}, X_t] + b_i)$, the forget gate $f_t = \sigma(W_f \cdot [H_{t-1}, X_t] + b_f)$, the candidate cell state $\tilde{C}_t = \tanh(W_C \cdot [H_{t-1}, X_t] + b_C)$, the updated memory cell state $C_t = f_t \cdot C_{t-1} + i_t \cdot \tilde{C}_t$, the output gate $o_t = \sigma(W_o \cdot [H_{t-1}, X_t] + b_o)$, and the hidden state $H_t = o_t \cdot \tanh(C_t)$. In the training process, these parameters $(W_f, W_i, W_C, W_o, b_f, b_i, b_C, b_o)$ are adjusted to reduce a preferred loss function, of course, using backpropagation over time.

LSTMs have been extraordinarily successful in several time series data analysis tasks, including natural language processing, speech recognition, weather forecasting, industrial process forecasting, and financial forecasting. LSTMs are enabled by their gating mechanisms to use and carefully update information from earlier time steps, making them particularly well suited to modeling sequential time series data with complicated dependencies.

## Convolutional neural networks

In particular, CNNs as displayed in Fig. 8, designed for image processing problems have been revised for time-series analysis by incorporating one-dimensional convolutions. This design revision allows CNNs to capture hierarchies and local patterns within sequential time series data. We have also tried to explain how CNNs work on sequential time series

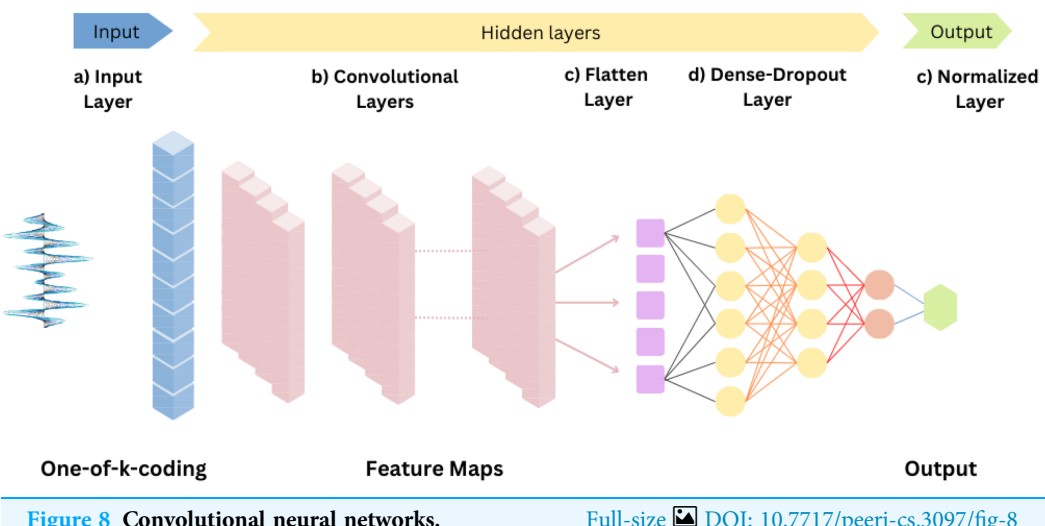

**Figure 8 Convolutional neural networks.**

data, including the mathematical underpinnings. In the case where we need to analyze time series data, CNNs use their one-dimensional convolutions to distinguish features or patterns within the sequential data. In the convolution layers, filters or kernels are applied to the input sequence of their local segments, capturing relevant patterns, called feature maps.

Suppose that X is the input time series data of length L, and the number of applied filters is K. The $i^{th}$ filter at position j, which is the convolution operation, is given by $(X^*W_i)_j = \sum_{k=0}^{M-1} X_{j+k}, W_{i,k} + b_i$. Where $X_{j+k}$ is the value of this input sequence at position $j + k$, the weight of the filter is $W_{i,k}$ for position k, for the $i^{th}$ filter $b_i$ is the bias term and the variable M represents the filter size. The result of the convolution operation is a feature map for each filter individually. After this process, an activation function, mostly ReLU, is applied element-wise to familiarize with the nonlinearity: ReLU $(X^*W_i + b_i)$. The reason for this is to learn more complex patterns. To reduce the computational load and also to reduce the dimensionality, *i.e.*, to down-sample the extracted feature maps, pooling layers are widely used. Within a certain window, this is MaxPooling $(X) = \max(X)$, which shows that the maximum or average value is taken by the pooling operation. Further fully connected layers can be added for advanced abstraction and prediction after one or more pooling and convolutional layers have been run. The result is classically convolved over one or more fully connected layers, followed by an activation function as required by the task. In addition, for effective training and learning, the network uses backpropagation and various hyperparameter optimizations so that the biases and weights are carefully tuned. 1D CNNs for time series data offer the convenience of dropping dimensionality, learning hierarchical representations, and capturing local patterns. 1D-CNNs have been effectively applied in various domains, including weather forecasting, health monitoring, financial prediction, and industrial process systems.

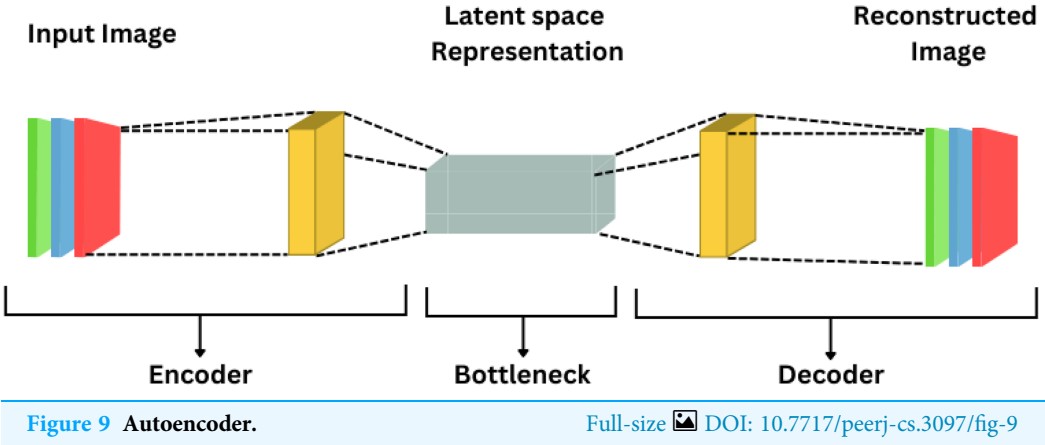

**Figure 9 Autoencoder.**

## Autoencoders

Unsupervised feedforward neural networks are represented as autoencoders shown in Fig. 9, designed with the goal of having their output closely match the input data. This architecture consists of encoder and decoder steps, where the encoder transforms the input information into a latent representation and the decoder reconstructs the input based on this representation. Typically, gradient descent methods are used to optimize the hyperparameters of the methods to minimize the reconstruction error. Notable alternatives to AEs include sparse autoencoders and denoising autoencoders.

## Transformers

The dominance of the transformer architecture has emerged as a spotlight in various domains such as computer vision tasks, natural language processing tasks, but their employment is not limited to only text or image data. Transformers also play and have proven competent in the field of time series, presenting an attention-based and parallelized style. We will briefly discuss the basic and fundamental approaches of transformers. Self-attention mechanism (Fig. 10): A mechanism that allows each component in the input sequence to emphasize additional elements in order to capture dependencies regardless of their locality. Given an input sequence $X = \{x_1, x_2, \ldots, x_n\}$, the self-attention score $e_{ij}$ between elements $x_i$ and $x_j$ is calculated as follows $e_{ij} = \text{softmax}(d_k(W_q x_i)^T(W_k x_j))$. Where $W_q$ and $W_k$ are learned weight matrices for queries and keys, and dk is the dimensionality of the key vectors. Scaled dot product attention: A weighted sum of the values ($W_v x_j$) calculated by the self-attention scores is used to obtain the output for each position. The weighted sum for each item $i$ is calculated as follows $\text{Attention}(Q, K, V) = \text{softmax}(d_k QK^T)V$.

Multi-head attention (Fig. 11): To increase the capacity of the model, multi-head attention uses multiple sets of learned linear projections in parallel. Let denote the number of heads. The output of each head is concatenated and linearly transformed: MultiHead(Q, K, V) = Concat(head$_1$, head$_2$, …,head$_h$), $W_o$ Here, $W_o$ is another learned weight matrix.

Encoder and decoder structure (Fig. 12): A transformer typically consists of an encoder and a decoder. The encoder processes the input sequence, and the decoder generates the

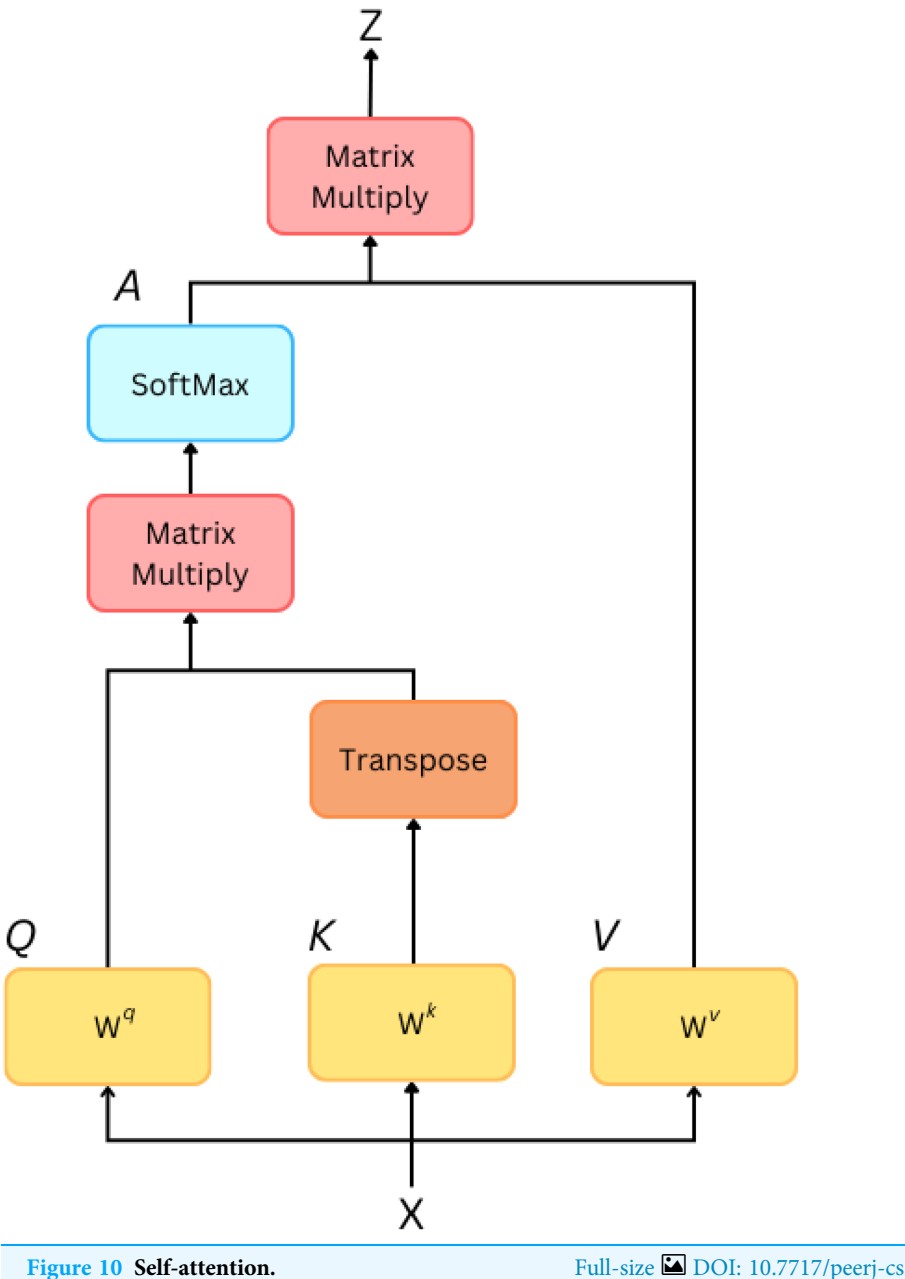

**Figure 10 Self-attention.**

output sequence. The encoder consists of several layers, each of which contains a multihead self-attention mechanism followed by feedforward neural networks. EncoderLayer(X) = FFN(MultiHead(X)). The decoder has an additional multi-head attention layer that monitors the output of the encoder and allows it to focus on relevant parts of the input sequence. DecoderLayer(X, EncoderOutput) = FFN(MultiHead(X) + MultiHead(X, EncoderOutput)). These basics set the stage for understanding how transformers can be applied to time series data. In particular, the self-attention mechanism allows the model to consider global dependencies and capture long-range relationships, making it suitable for analyzing sequential information.

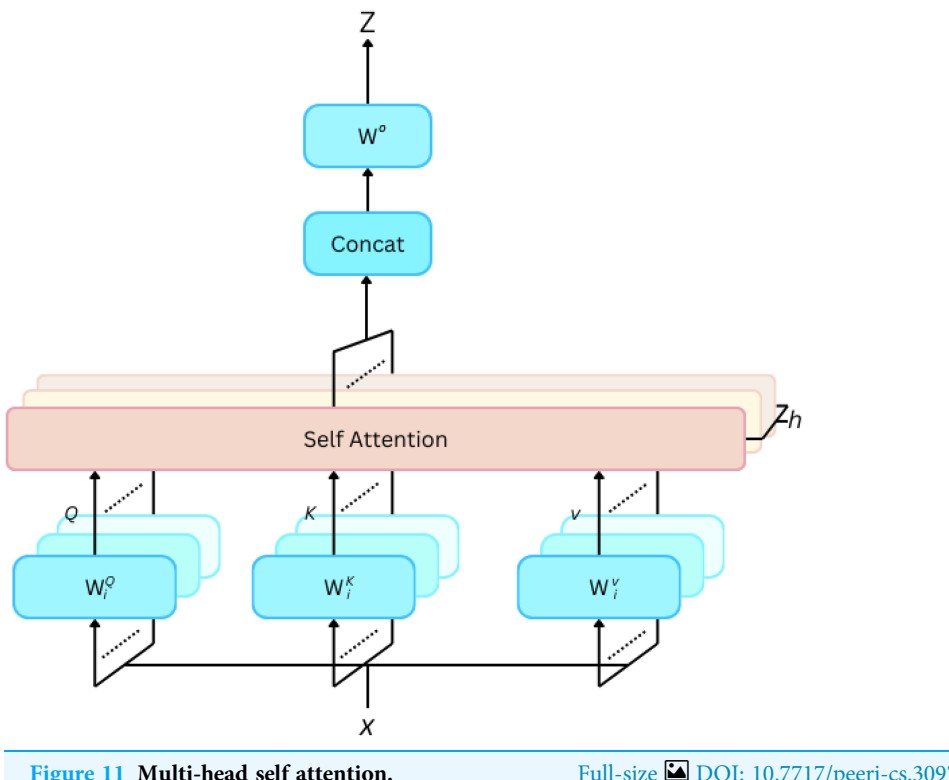

**Figure 11 Multi-head self attention.**

## MACHINE LEARNING: INSIGHTS INTO FORECASTING, ANOMALY DETECTION, IMPUTATION, AND CLASSIFICATION STRATEGIES

Exploring the multifaceted field of time series analysis, this section delves into various works that shed light on prediction, anomaly detection, and classification. By reviewing a spectrum of methodologies, we unravel the intricacies of temporal data and provide a comprehensive view of advances in predictive modeling, anomaly detection, and event classification within the dynamic landscape of time series analysis. Accurate forecasting is imperative for improving industrial actions, moving towards machine learning, and two models taking dominance, Autoregressive Integrated Moving Average (ARIMA) and LSTM network techniques, play a critical and significant role. The study of *Xu et al. (2022)* begins a reasonable investigation of ARIMA and LSTM methods, exfoliating light on the model's advantages and limitations in prediction accuracy and computational competence. LSTM prove effective due to their ability to capture long-term dependencies in circumstances where future outcomes or predictions are significantly influenced by historical ones. However, it is expected that limitations may arise which can lead to overfitting, particularly when training with very inadequate data. Although techniques such as ARIMA are suitable for capturing linear dependencies, a situation may arise where they fail to adapt quickly to unexpected changes. In the research of *Binkowski, Marti & Donnat (2018)*, the researchers investigate the analogy of ARIMA and LSTM methods, highlighting the volume of LSTMs to capture nonlinear dependencies. This study commits

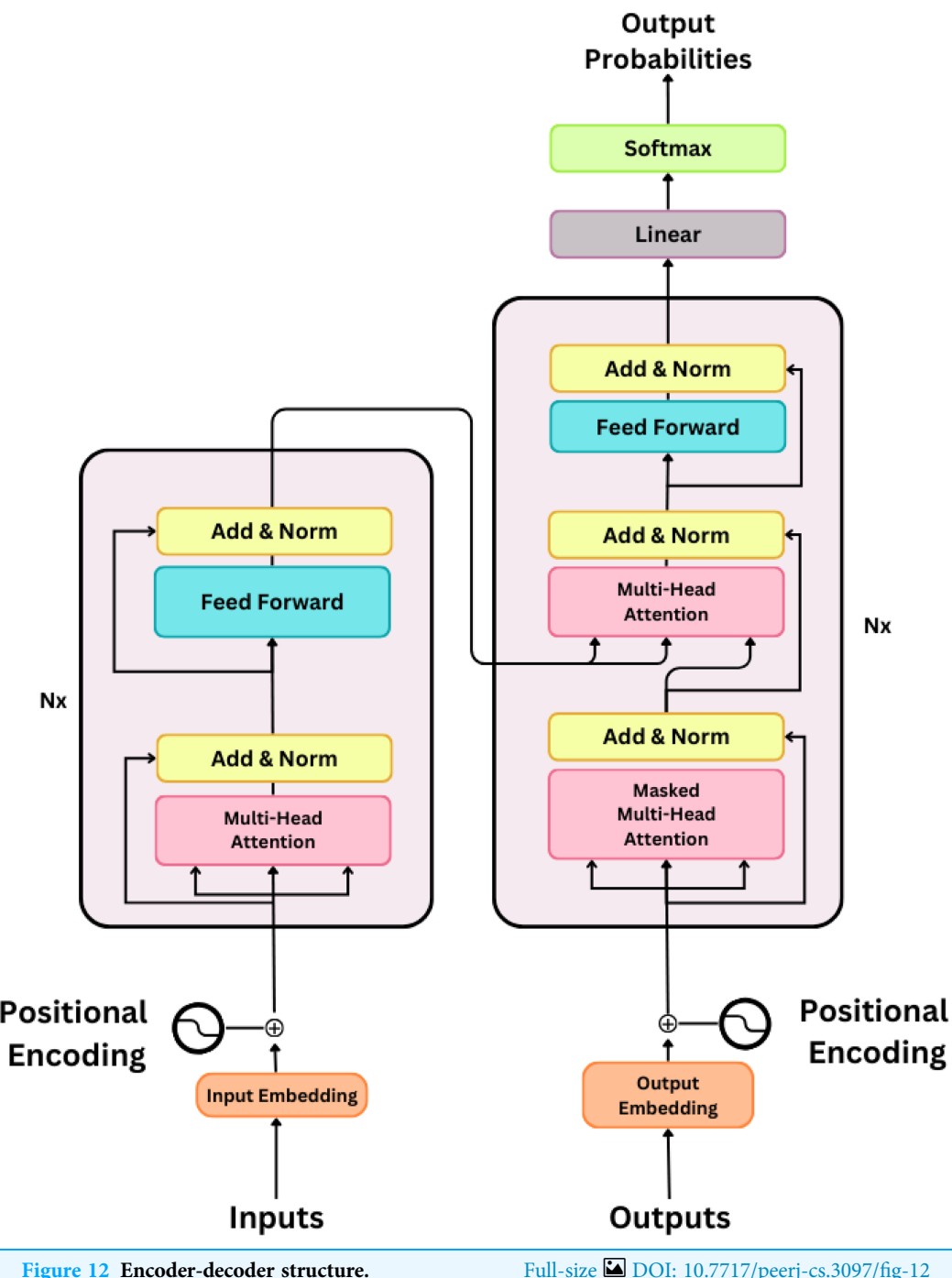

**Figure 12  Encoder-decoder structure.**               

to intuitions in the choice of appropriate techniques based on data characteristics. *Hewage et al. (2020)*, proposed temporal convolutional networks (TCNs), exploring the functions of TCNs to demonstrate their success in capturing long-term dependencies, increasing the accuracy and efficiency to predict time series in industrial progressions.

Techniques such as ensemble learning are further explored in *Du (2019)*. The study discusses the prediction of wind power generation using ensemble methods and highlights

the advantages of combining several techniques to improve accuracy. Probabilistic prediction is an essential prerequisite, and the study by *Pinson (2013)*, emphasizes dependent (conditional) autoregressive methods for probabilistic prediction. This study offers insights into how to deal with uncertainty in prediction, especially in the context of renewable energy. In industrial time series data, the detection of anomalies is very important and critical for avoiding machine failures and ensuring system safety. A study conducted by *Jiang et al. (2021)* educates us on anomaly detection which is a very beneficial insight. The research discovers the use of SVM and isolation forests to detect outliers. SVM, which excels in cases where there are unbalanced classes, nevertheless requires great care in hyperparameter tuning. On the other hand, isolation forests, which take advantage of the isolation principle, show competence in training large data sets, making them suitable and favorable for industrial situations. However, limitations and threats arise when outliers are deeply rooted in condensed clusters. The research of *Ullah & Mahmoud (2021)* extends the argument to use different algorithms for anomaly detection in sensor data in the industrial Internet of Things (IIoT) environment.

The study provides a proportional analysis and comparison of algorithmic performance in the context of industrial IoT. Another study by *Schubert, Gupta & Wever (2023)* explores scalable anomaly detection. They develop techniques that are scalable for anomaly detection, keeping in mind the suitability for large dynamic systems, emphasizing the adaptability in handling growing anomalies in industrial developments. *Olivieri, Colleoni & Bonaccorso (2023)* argues on the impression of developing tendencies in predictive analytics in Industry 4.0, also taking into account anomaly detection. While she is more focused on the survey of tendencies, she sheds light on the insights of integrating machine learning models for real-time anomaly detection. Regarding streaming data, *Simkute et al. (2021)* discusses techniques for real-time anomaly detection. It also explores approaches that are best suited to streaming data, a critical requirement in industrial environments. In the classification orbit, machine learning methods such as neural networks, k-Nearest Neighbors (k-NN), and Random Forests dominate as key players. *Fahim & Sillitti (2019)* shares his thoughts on Random Forests, highlighting its efficiency in determining valued perceptions and educating decision making. The robustness of methods such as Random Forests, as opposed to overfitting, demonstrates their suitability for scenarios and claims where consideration of the decision-making process is very important. However, due to their ease and simplicity, they may struggle to capture complicated relationships in complicated time series data. *Shrestha, Krishna & von Krogh (2021)* explore k-NN precisely for the classification domain in time series. They highlighted the success of k-NN in capturing local patterns, assembling it to be a valued tool in very dynamic environments of the industry. However, we keep in mind the high dimensionality of time series data, which will have its challenges and will require careful attention. *Wang et al. (2021b)* sheds light on different classification models in the time series domain, which include neural networks, k-Nearest Neighbors and Random Forests. The investigation provides insight into the advantages and disadvantages of different approaches. In investigating the classification domain in time series with respect to machine learning methods, *Guigou (2019)* provides an overview that summarizes the

growing landscape of machine learning models for classification in the context of time series. The comparative analysis of *Wang et al. (2022)* enhances the perception of classification in time series data.

## ADVANTAGES AND LIMITATIONS OF MACHINE LEARNING MODELS IN INDUSTRIAL TIME SERIES ANALYSIS

In the domain of industrial time series investigation, machine learning techniques suggest several advantages, but they contend with integral limitations. With considerable advantages, machine learning models in the industrial time series domain, it is inescapable to endorse their limitations and constraints. *Idowu, Strüber & Berger (2021)* The scalability of ML techniques is discussed in detail; they provide useful insights into the application of ML at scale, focusing on the benefits of appropriate handling of large industrial datasets using distributed computing. However, there is the possibility that, considering the technological essence of scalability, constructive adaptation may be required for explicit industrial practice. Speaking of the interpretability of ML methods, *Zhang et al.*'s *(2022b)* research highlights the importance of constructing black-box methods that are interpretable. Although it provides direction on how to achieve interpretability, the hypothetical nature of the argument may require a real-world transformation for industrial frameworks. Furthermore, the *Shalev-Shwartz, Shammah & Shashua (2018)* investigation of generalization highlights encounters with generalisation in industrial settings. They also impart practices to promote model robustness, highlighting the need for techniques to adapt to different operating environments.

However, the hypothetical nature of the argument requires practical investigation for certain industrial settings. The ethical considerations associated with the production and use of ML methods are addressed in *Jaton (2021)*. The research discovers the possible social effects of automation and artificial intelligence in industrial settings, highlighting the importance of accountable machine learning protocols. While the emphasis on ethical considerations may be crucial, it may require additional research for practical implementation in various industrial settings. Adapting to the evolving nature of industrial operations is one of the essential qualities of the volume of ML methods, allowing immediate integration of changes in data arrangements (*Tseng et al., 2021*). This conformance is particularly valuable in industries where operational situations change vigorously, polishing the competence of models to maintain accuracy in the seam of changing conditions. However, realizing the generalizability of these robust techniques across diverse industrial settings requires additional investigation. ML methods have established substantial success in predicting sustainable use within industrial operations (*Koste & Malhotra, 1999*). By examining time series or sequential data, these techniques can predict machine deterioration, enabling positive maintenance interventions and limiting unexpected breakdowns.

The predictive skills of ML promise to extend the maintenance route, thus significantly improving functional productivity. On the other hand, the practical application of predictive maintenance approaches requires meticulous validation in contrast to concrete machine settings to guarantee effectiveness and reliability. Furthermore, the combination

of ML techniques promises to improve automation and optimize complex processes in industrial control systems (*Jiang et al., 2021*). These techniques, skilled enough to adapt data patterns and learn these patterns, are likely to advance real-time decision-making and asset allocation efficiency. Nevertheless, the placement of ML in control systems requires an exhaustive inspection of the consistency and safety of the inferences, since in industrial frameworks any failure could have unadorned penalties. Regardless of these advantages, in industrial frameworks, time series analysis lacks the interpretability of ML techniques and its residue is a substantial encounter (*Carvalho, Pereira & Cardoso, 2019*).

The intrinsic complication of certain algorithms can obscure their decision processes, making it difficult to extract meaningful information from time-series data. Addressing this encounter requires advanced methods to improve the interpretability of these techniques without compromising their predictive accuracy. Furthermore, the energy competence of these methods in environments where the industry is resource constrained is a significant concern (*Aldhaheri et al., 2024*). Deploying and training skeptic ML methods requires extensive computational power, and ensuring the energy competence of these methods is imperative for sustainable industrial operations. Discovering optimization strategies and energy-efficient ML designs is the key to mitigating the environmental burden of extensive ML implementation in industrial frameworks. These techniques offer considerable support in the analysis of industrial time series, but their implementation is not free from limitations and restrictions.

The consideration of unbalanced data sets, trivial in industrial settings, is crucial to avoid biased methods. *Qian, Vaddiraju & Khan (2023)* Focal points to the obligation of developing models to grasp imbalances occurring in data classes, guaranteeing reliable technique results. An important concern that poses adversarial attacks, particularly in dangerous industrial arrangements (*Aryal, Gupta & Abdelsalam, 2022*). Shielding ML techniques from premeditated data handling involves vigorous security benchmarks that protect decision processes persuaded by model productivities. The resource-intensive nature of the training of complicated machine learning techniques presents a threat to smaller industrial units with inadequate computational frameworks (*Abdeldayem et al., 2022*). Regularizing the implementation of machine learning in different industrial settings requires the advancement of resource-efficient training approaches and lightweight model architectures. Interpretability remains a crucial challenge, especially in applications that are very critical in safety critical aspects where the decision-making method is crucial (*Papenmeier, Englebienne & Seifert, 2019*). The generation of interpretable and transparent machine learning techniques, customized to industrial requirements, is crucial for fostering trust and facilitating operational decision making.

In ML implementation, ethical considerations deserve careful consideration, as emphasized by *Boardman (2022)*. Talking of socio-economic impacts and biases in ML education requires the expansion of explicit ethical contexts for the use of industrial ML (*Boddington, 2023*). Exploring the multifaceted sociotechnical perspective is significant for supporting responsible AI protocols in the industrial sector. In conclusion, while ML techniques hold excessive potential for analyzing time series in industrial environments, certifying security, mitigating imbalance, regularizing access, reviving interpretability, and

acquainting with ethical issues are dynamic steps towards their active and responsible implementation in industrial frameworks.

# DEEP LEARNING: INSIGHTS INTO FORECASTING, ANOMALY DETECTION, IMPUTATION, AND CLASSIFICATION STRATEGIES

Starting from the pioneering frontier of time series investigation, this segment explores the competence of deep learning techniques in reshaping the field of prediction, anomaly detection, and classification. From extracting multifaceted temporal patterns to improving interpretability, these cutting-edge methods demonstrate the changing latent of deep learning across diverse applications within the domain of time series analysis. The increasing diffusion of technology has led to a gradual intensification in the use of soft measurement techniques for monitoring and analyzing industrial systems. The complexity inherent in thermal-hydraulic processes, heat-mass transfer, and physicochemical reactions within various industrial frameworks, including those in the petroleum, nuclear, thermal, chemical, and power sectors, poses challenges in establishing highly accurate mechanism models for predicting operational states (*Kadlec, Gabrys & Strandt, 2009*). Therefore, unlike methods based on mechanism models, time series prediction using a data-driven approach proves to be more appropriate and offers improved development and applicability to process systems within the industrial sector (*Kano & Ogawa, 2010*; *Karl Pearson, 1901*).

Recently, the term deep learning has been applied to methods that use different layers to capture latent features at advanced and more abstract levels. Deep learning attempts to select complex abstract features employing both non-linear and linear changes, providing a viable approach to address this problem. In the historical literature, four typical deep learning models have been defined and are widely used in time series forecasting tasks in many process systems in industrial environments. These models include deep belief networks, recurrent neural networks, encoder-decoder, and convolutional neural networks. These well-known models have shown successful applications in a variety of predictive or forecasting tasks. Recurrent neural networks stand out as a popular choice for achieving high performance in forecasting and prediction tasks. Examples include prediction of remaining useful life (RUL) for aircraft engines designed by *Yan, Shao & Wang (2004)*, prediction of solar power level proposed by *Miao et al. (2019)*, prediction of soft sensor industrial data established by *Wang et al. (2024)*, and a system to predict the temperature of long-term steam generator (SG) in a nuclear facility reported by *Yuan et al. (2020)*.

In addition, some scholars are trying to model hybrid methods, either in their unique forms or different forms, to effectively address the forecasting task. For example, *Nguyen, Liu & Zio (2020)* designed an approach that combines a variational recurrent unit with a Bayesian network in an IoT framework to achieve time series forecasting in meteorological data (*Das & Ghosh, 2019*). *Wang et al. (2021a)* proposed a CNN-LSTM network as a solution to forecast the levels in electrical power structures (*Zheng et al., 2023*). *Zheng et al. (2023)* used a hybrid CNN-LSTM model to capture 2D

link information and retain long-term memory to predict different traffic scenarios. *Lima et al. (2021)* integrated LSTM-CNN with an autoencoder to predict time series in metal packaging plants. Another study (*Zhang et al., 2023*) used principal component analysis for the direct extraction of multivariate time series features and combined it with a CNN-GRU method, which is again a hybrid model. A study by *Wang et al. (2020a)* introduced the variational mode decomposition integrated with a GRU technique. Furthermore, *Yu et al. (2022a)* supported a method for predicting wind speed using an LSTM-based method with empirical wavelet transform. Recently, there has been a growing passion and interest in deep neural networks (*Zintgraf et al., 2017*), driven by numerous demonstrated successes in various tasks (*Jain, Nandakumar & Ross, 2016*).

The proven ability of these methods to detect advanced correlations in complex data, often characterized by significant volume and dimensionality, contributes to their popularity (*Vakaruk et al., 2021*). This trend has not spared anomaly detection from multivariate time series data, which has witnessed a surge in DNN-based methods. These methods have proposed systematic developments and demonstrated improved performance, as highlighted in *Rana et al. (2020)*. Methods based on deep neural networks seek to acquire deep latent representations of time series in multivariate data, allowing the inference of a model of variability. This learned model is then used to classify anomalies in previously unseen data. The surge in the adoption of deep neural network architectures is driven by the need to understand potentially complex data patterns underlying the temporal evolution of multivariate time series data. Building on the aforementioned rationale and inspired by the laudable achievements of DNNs in various domains, researchers have shifted their focus away from direct contrasts with other traditional approaches, such as conventional techniques or statistical methods, including machine learning (*Avci et al., 2021*).

This tendency has motivated researchers to develop increasingly complex models to improve the performance of DNN-based models. However, this is often done without substantial theoretical or empirical evidence demonstrating their superiority over more established techniques documented in research (*Vakaruk et al., 2023*). Training DNN-based techniques is a complex process that involves the evaluation of a significant number of hyperparameters, requiring large training model sizes and significant computational characteristics. Furthermore, the complexity of these models continues to increase with the ongoing development of larger and more complex architectures. On the contrary, traditional models tend to be simpler, easier to understand, lighter in weight, and often well adapted to the practical constraints of real-world applications.

Therefore, it is essential to determine whether the complication introduced by deep neural network methods is an essential trade-off for performance gains, or whether the advances reported in recent years are misleading (*Al-Garadi et al., 2020*) and advocate a preference for traditional methods. Several studies have investigated the actual benefits of DNN-based methods in various application domains. An initial study by *Jiao et al. (2020)* demonstrated that methods using conventional linear regression techniques outperformed methods based on deep neural networks in blind reconstruction in single-pixel imaging and an optical cryptosystem outbreak. *Zhang et al. (2018)* introduced a constancy test and

showed that image reconstruction methods based on deep neural networks have a high sensitivity to small perturbations in the input images during training, leading to unbalanced results.

In addition, *Maron et al. (2021)* has shown that even small changes in the input to a deep neural network, often imperceptible to an individual, can disrupt even the most proficient neural networks. This underscores the lack of robustness of methods based on deep neural networks and the fact that these models rely heavily on large amounts of data. The concept of a digital factory envisages a highly digitized and networked integration of equipment and machinery, designed to increase production and value through self-optimization and automation. In a computerized industrial process, equipment settings are closely linked to both productivity and quality. A stable process contributes to an increase in value, while an efficient process reduces assembly time and supports production. Therefore, the timely detection of faults or the anticipation of potential anomalies in the equipment is critical. The equipment used in digital factories includes manufacturing equipment, structure equipment, and logistics digitization equipment.

Manufacturing equipment is responsible for ensuring that products are manufactured efficiently and to excellent standards. Meanwhile, infrastructure plays a key role in supplying gas, water, chemicals, and electricity to the manufacturing system. It is also involved in the management of chemical waste and wastewater treatment. Logistics digitalization facilities are tasked with transporting goods from the point of origin to the other point. Although various machine learning methods are used to identify anomalies, faults, and damage in such industrial systems (*Fernandes, Corchado & Marreiros, 2022*), deep learning models have shown significant potential in this regard. Data-driven models play a key role in improving system operation in large manufacturing industries, as they have the ability to identify potential failures without requiring extensive knowledge of the domain. *Chevrot, Vernotte & Legeard (2022)* used an LSTM-based autoencoder to study the standard state of the equipment and identify anomalies in various multivariate time series streams related to production equipment machines. The LSTM-based autoencoder consists of an encoder and a decoder, each of which incorporates long- and short-term memory networks, which are modifications of recurrent neural networks. Following the revolutionary impact of convolutional neural networks in computer vision (*Smith, Smith & Hansen, 2021*), scientists have extended their application to the analysis of time series data (*Shumway & Stoffer, 2017*). CNN-based diagnostic and fault detection models have demonstrated their effectiveness in managing multivariate time series data from semiconductor manufacturing developments in studies such as *Hsu & Liu (2021)*, *Zhu et al. (2021)*.

## ADVANTAGES AND LIMITATIONS OF DEEP NEURAL NETWORKS

In the dominion of time series investigation, multifarious practices and methods address definite challenges. A study *Gaugel & Reichert (2023b)* proposes an efficient labeling technique for time series, using a segmental strategy with three dedicated modules for probable performance developments. The challenges continue in rapid fashion

engineering, where a deep learning architecture in *Wang et al. (2021b)* integrates visual, time series, and textual data to improve the detection of different trends in the fashion industry, regardless of the constraints imposed by the lack of historical data. For fault diagnosis, a technique integrating LSTM and the residual convolutional neural network validates the efficiency of extracting characteristics from time series data with compact training effort (*Yao, Yang & Li, 2021*).

Challenges include understanding the details of the model, a lack of data to evaluate, and the effect of small data proportions. In industry and engineering, an optimization and machine learning method highlights waste reduction and resource savings, encountering limited set of certain instances, and data accessibility complications (*Dambros et al., 2019*). Feature extraction based on scalable hypothesis tests (*Weichert et al., 2019*) presents feature extraction with great efficiency, integrating metainformation with parallelization and time series. It lacks to further address the prerequisite of domain knowledge and also very incomplete points on evaluation metrics. Another study (*Christ, Kempa-Liehr & Feindt, 2017*), proposes a feature selection based on principal component analysis, confronted with extensive execution work and a lack of standard datasets.

Reasonable research in anomaly detection techniques for industrial control systems highlights the effect of size set in training, illustrates implementation variances, and reveals limitations in method diversity. Some studies such as *Wang, Yang & Li (2020)*, which introduces how to form an image for signals, also illuminate practical applications, but weaken in the discussion of network architecture and in-depth comparative analysis. An approach (*Tziolas et al., 2022*) for unbalanced data using a GAN and feature extractor limits the argument on the in-depth description and research on hyperparameter tuning details and how they built the feature extractor. *Liu, Hsaio & Tu (2019)* introduces process optimization, a tensor structure for ensemble modeling, and data transformation. Constraints arise from the installation of frequent techniques in production and a limited argument on unbalanced data encounter. Incorporating data analytics and sensors to improve process manufacturing, they propose an ARIMA-based predictive maintenance technique, deficient in detail of key metrics and speaking of security as well as privacy concerns. In tissue engineering, the combination of time series technique and dielectric spectroscopy demonstrate improved prediction with LSTM, but deficit precision on external validation, dataset size, and pertinence to longer-term value situations. *Kim et al. (2023)* presents Gated CNN-based Transformer, which demonstrates effective soft-sensor modeling but shortcomings in comprehensive analysis of trade-offs and component effects. *Liu et al. (2021)* proposes contrastive predictive coding (CPC) to label effectiveness but has limitations in real-world scenarios and comparisons.

*Jiang et al. (2019)* investigates multi-head CNN-RNN for anomaly detection, which uses independent CNNs for different sensors, but finds limitations in baseline comparison. *Zhang et al. (2021)* introduces a hybrid model of bidirectional deep inner partial least squares (DiPLS) with LSTM, which shows better developments in interpretability but requires careful consideration of the extended computational task. The transfer probability of Markov chains (*Kanawaday & Sane, 2017*), a feature extraction technique, expresses limitations in lack of intuition in information security, insufficient comparison debate, and

sensitivity to hyperparameters. The attention-based architecture (*Jatti, Sekhar & Shah, 2021*) for soft sensor simulation presents spatiotemporal attention, but has a deficit in exploring the model interpretability and sensitivity tasks in industrial frameworks. Giving manufacturing process data to a branched LSTM as a spatial sequence investigates fault detection, but limitations arise in terms of tuning hyperparameters essentials applicability. Transfer learning setups for time series partitioning provide insights into training speed, but deep investigation of layer-frozen residues is a challenge. *Ali Nemer et al. (2022)* introduces a linear-discriminative generative-discriminative method, this technique competently discovers deep neural network pipelines, highlighting productivity over precision. Limitations consist of the trade-off between accuracy and efficiency, limited validation, and domain specificity. *Geng et al. (2022)* presents a TC-GATN architecture that employs temporal causal graph attention networks that engage nonlinear relationships, which also faces challenges in dependence on causality information and complexity.

The fused techniques of *Gamage, Klopper & Samarabandu (2022)* integrate XGBoost and GRU, to extract features and timing knowledge, highlighting key feature outputs but demanding additional investigation of security and scalability fears. A method of *Schockaert (2020)* employing continuous learning reports a nonstationary challenge in data distribution, but has deficiency in strict evaluation and neglects resource and privacy concerns. A study by *Fährmann et al. (2022)* introduces a segmental data driven system to predict failures in different machines, it adapts to variations but lacks on how to handle data correctness, class imbalance, and lack of feature facts. *Canizo et al. (2019)* propose a hybrid prototype for anomaly detection in industrial IoT frameworks, which emphasizes real-time intuitions but requires additional investigation of privacy and scalability issues. The research by *Wang, Bao & Qin (2023)* presents a technique to predict multistep time series data in industrial processes, proposing a self-motivated transfer learning architecture that integrates multitasking and domain adaptation. However, there is a clear lack of detailed elucidation of a number of workings, including neural network construction and hidden layer selection principles, which limits the robustness of the technique. Limiting generalization to real-world industrial settings with a dataset of 2,859 samples also raises questions about model scalability. A new method by *Zang, Liu & Wang (2018)*, for the sole purpose of maintenance prediction in the case of industrial drying hoppers, uses a deep learning technique that targets unbalanced data in time series scenarios. However, the lack of a physics-based prototype poses challenges in terms of interpretability. Limitations in accurately characterizing abnormal procedures and bias in event classification influence the reliability of the technique in various industrial settings.

The use of generative techniques to integrate time series with image coding for fault detection is an innovative method. But it faces challenges in reconstructing the image and obsessiveness with data size, complementing concerns about versatility in different industrial settings. Neural network using Bayesian knowledge introduced by *Yuan et al. (2021)*, which is a stochastic device to detect anomaly refrigeration engineering. Uncertainty actions for curative activities depend on inadequate labor comfort, questioning practical relevance in real industrial setups. A deep network-based hybrid

architecture of *Li (2021)*, for short-term multivariate time series prediction in the industrial IoT setting, merges several mechanisms. However, its precise suitability to the Industrial IoT framework and limited insight into selected features affect its interpretability. The preliminary hyperparameter service and its reworking by means of Bayesian optimization license additional investigation (*Gaugel & Reichert, 2023a*). The method for fault detection using 1D CNNs (*Shojaee et al., 2021*) in time series scenarios adapted to architectural improvements. Lack of highlighting how to handle a small training size, balancing fault event intricacy, overfitting concerns with appropriate sequence length for flawless performance.

## GRAPH NEURAL NETWORKS: INSIGHTS INTO FORECASTING, ANOMALY DETECTION, IMPUTATION, AND CLASSIFICATION STRATEGIES

As we navigate the complicated realm of time series, this unit sheds light on the contributory character of GNNs in transforming forecasting, anomaly detection, and classification. From interpreting complicated dependencies within systems that are highly interconnected, to demonstrating an all-encompassing use of temporal information, GNNs position themselves at the forefront of modernization, restructuring the changing aspects of time series submissions through their exceptional graph-oriented method. Time-series forecasting involves predicting future values in a time sequence by relying on past observations. The roots of time series forecasting can be identified in models (*Li et al., 2018*) that are statistically autoregressive. Their sole purpose is to use a linear combination of past values within a time series to predict its future values. In recent years, deep learning-based approaches have shown significant efficacy in predicting time series results, demonstrating a superior ability to capture non-linear temporal and spatial patterns (*Gao & Ribeiro, 2022*). Various methods have been used for time series prediction, including attention-based neural networks, CNNs, and RNNs. However, several approaches, including LSTNet (*Pan et al., 2019*) and temporal pattern attention-long short-term memory (TPA-LSTM) (*Zambon et al., 2019*), indirectly model or tend to overlook the complex dynamic spatial correlations present in time series. To effectively and explicitly model the spatial and temporal dependencies within the multivariate time series data, there have been notable advances in the use of methods based on GNNs. This has led to improved forecast performance. Forecasting models based on GNNs can be categorized and studied from different perspectives (*Ahmed et al., 2021*). In terms of forecasting tasks, many models focus on multistep forecasting—predicting several successive future steps using historical data.

In contrast, a minority deals with single-step forecasting, *i.e.*, predicting only a single or next arbitrary step ahead. The analytical dissection of these time series forecasting models involves three perspectives: exploring spatial (*i.e.*, intervariable) dependencies, examining intertemporal dependencies, and architecturally integrating temporal and spatial modules. The relationships between spatial elements, also known as intertime series connections, have a significant impact on a model's ability to forecast (*Guo et al., 2019*). Some of the methods such as (1) spatial GNNs, (2) spectral GNNs, or (3) a fusion of both are

commonly used by contemporary research in some cases where they are faced with accompanying graph structures and time series data that can be used to illustrate the strength of connections in them to model and capture these spatial dependencies in time series. In the early stages, GNN-based forecasting models mainly used ChebConv (*Wu et al., 2019*) to approximate graph convolution using Chebyshev polynomials, effectively modeling and capturing dependencies between time series. As an example, ChebConv layers and temporal convolution were combined to effectively capture temporal and spatial patterns. Proposing temporal and spectral graph neural networks, StemGNN (*Chen et al., 2020*) aims to use the frequency domain and ChebConv convolutional neural networks to extract complex patterns from time series.

Subsequent relevant studies have mostly followed this approach, using ChebConv to introduce inventive modifications and to model spatial time-series dependencies. Such adaptations include multigraph construction (*Wang et al., 2020b*; *Zheng et al., 2020*), attention mechanisms (*Zhang et al., 2020a*; *Guo et al., 2019*), and hybrid combinations of the two (*Yu et al., 2020*). Based on StemGNN, theoretical evidence has recently been provided by *Paassen et al. (2020)* that demonstrates the advantages of using spectral GNNs to model different signed time series relationships. This includes variables with strong negative and positive correlations in a multivariate time series. It was also found that for such tasks, any orthonormal family of polynomials can achieve comparable expressiveness, albeit with different empirical performance and convergence rates. Inspired by the recent successes of spatial GNNs (*Chen, Segovia & Gel, 2021*), an alternative line of research model dependencies between time series through graph diffusion (*Lan et al., 2022*; *Liu et al., 2022*) or message passing (*Choi et al., 2022*). Looking at the graph, it becomes clear that these approaches are specific simplifications in contrast to those based on spectral GNNs, which highlight robust local homophilic tendencies (*Shao et al., 2022*; *Chauhan et al., 2022*). In early techniques such as DCRNN (*Yu et al., 2022b*) and Graph WaveNet (*Cui et al., 2021*), GRU (*Zhao et al., 2020*), temporal convolution or graph diffusion layers are integrated to effectively capture patterns in time series data. In later studies, graph diffusion has been used by some approaches such as Spatio-Temporal Graph Diffusion Network (ST-GDN) (*Deng & Hooi, 2021*) and graph time series (GTS) (*Grattarola et al., 2020*). On the contrary, spatio-temporal graph convolutional networks (STGCN) (1st), which is a later iteration of ST-MetaNet (*Dai & Chen, 2022*) and STGCN (*Zambon, Livi & Alippi, 2022*), adopted graph convolutional network (GCN) (*Han & Woo, 2022*) and graph attention network (GAT) (*Chen et al., 2022c*) to characterize spatial dependencies, allowing the aggregation of information from neighboring time series. Other relevant studies, including multi-range attentive bicomponent GCN (MRA-BGCN) (*Wu et al., 2021*), Spatio-Temporal Graph Neural Networks (STGNN) (*Chen et al., 2022a*), graph multi-attention network (GMAN) (*Guan et al., 2022*) and adaptive graph convolutional recurrent network (AGCRN) (*Srinivas, Kumar Sarkar & Runkana, 2022*), introduced adaptations to capture relationships between time series through message passing. The spatial-temporal synchronous graph convolution was introduced by Spatio-Temporal Synchronous Graph Convolutional Network (STSGCN) (*Chen, Feng & Wirjanto, 2023*) to improve learning efficiency.

This extension of GCN is specifically designed to capture both spatial and temporal dependencies within localized spatio-temporal graphs. Before using graphs and temporal convolutions, Spatio-Temporal Fusion Graph Neural Network (STFGNN) (*Chen et al., 2022b*) formed fusion graphs that integrated spatial and temporal aspects using dynamic time wrapping (DTW). Zero-Inflated Graph Convolutional Networks (Z-GCNETs) (*Zhou, Zeng & Li, 2022*) improved current approaches by integrating prominent time-dependent topological data, with particular emphasis on zigzag persistence images. Multiscale temporal graphs were introduced by METRO (*Duan et al., 2022*) to represent dynamic temporal and spatial relationships within time series data. This was achieved by incorporating cross-scale graph fusion modules and single-scale graph updates, with the aim of integrating the modeling of spatial-temporal dependencies. An alternative avenue of extension is the integration of graph propagation, which allows the merging of substructures and higher-order relationships within the network. Examples of this include spatial graph propagation (SGP) (*Aldhaheri et al., 2024*) and multivariate time-series graph neural network (MTGNN) (*Chauhan et al., 2022*). Specifically, SGP uses multihop spatial processing and reservoir computing to generate precomputed spatio-temporal representations, resulting in efficient and scalable predictive models. Graph propagation and continuous graph propagation were introduced by subsequent research efforts such as multivariate time-series graph ordinary differential equation (MTGODE) (*Ao & Fayek, 2023*) and temporal pattern graph neural network (TPGNN) (*Srinivas, Kumar Sarkar & Runkana, 2022*) based on temporal polynomial coefficients. Comparable contributions in this area include spatio-temporal graph ordinary differential equation (STGODE) (*Chauhan et al., 2022*) and spatio-temporal graph neural controlled differential equation (STG-NCDE) (*Shao et al., 2022*).

Unlike GAT-centric approaches, methods based on graph transform (*Zambon, Livi & Alippi, 2022*) can capture extended spatial dependencies due to their extensive coverage, which puts them in a distinct category of improved techniques. The goal of time series anomaly detection is to detect data observations that deviate from the typical pattern established by the data generation process. Any data point that falls outside this pattern is characterized as an anomaly, while data points within the established pattern are referred to as normal data. It is worth noting that in the literature, terms such as novelty and outlier are often used almost interchangeably with the term anomaly. Deviations from standard conditions may manifest themselves as a single data point or as a series of observations. Unlike regular time-series data, anomalies present a challenge in characterization due to two main factors. Firstly, they tend to be associated with infrequent events, making the collection and labeling process difficult. Second, defining the full range of possible anomalous events is typically unattainable, compromising the effectiveness of supervised learning methods. As a result, unsupervised detection techniques have been extensively explored as a pragmatic approach to solving complex real-world problems. Historically, approaches such as distance-based methods and distributional techniques have been widely used to identify anomalies in time series data. Using distance metrics, the former category measures the deviation of observations from representative data points, while the latter focuses on identifying anomalies by targeting points with low probability. As the

complexity of the data generation process increases and the dimensionality of the multivariate time series expands, these approaches become less effective.

With the advancement of deep learning, initial studies have suggested the use of recurrent models that employ both reconstruction and prediction approaches for improved anomaly detection in multivariate time series data. Prediction and reconstruction strategies rely on prediction and reconstruction errors as measures of the discrepancy between expected and actual signals. The idea is that if a model trained on normal data has difficulty predicting or reconstructing certain data, there is a greater likelihood that such data are indicative of an anomaly. However, recurrent models (*Aldhaheri et al., 2024*) have been found to do a poor job in explicitly modeling pairwise interdependence between pairs of variables, thereby limiting their effectiveness in identifying complex anomalies (*Angelov & Gu, 2019*). Recently, GNNs have shown significant potential in bridging this gap by skillfully capturing both spatial and temporal dependencies between variable pairs (*Ali Nemer et al., 2022*). An unsupervised approach to anomaly detection requires models to develop a comprehensive understanding of the characteristics that define normality within a given dataset (*Canizo et al., 2019*). Advances in anomaly detection and diagnosis have led to the introduction of more comprehensive backbone and scoring modules (*Cheng et al., 2023*), largely influenced by the adoption of GNN methods (*Choi et al., 2022*).

The goal of time series classification is to assign a categorical label to a given time series, and this assignment is based on the inherent features or patterns within the time series. As highlighted in a recent review (*Ali Nemer et al., 2022*), the initial literature on time series classification focused mainly on distance-based methods for assigning class labels to time series (*Liu, Ting & Zhou, 2012*) and ensemble techniques such as hierarchical vote collective of transformation-based ensembles (HIVE-COTE) (*Lines, Taylor & Bagnall, 2018*). However, despite their state-of-the-art performance, both approaches face scalability limitations when applied to high-dimensional or large datasets. Exploring the possibilities of deep learning techniques, experts have initiated efforts to improve the effectiveness and scalability of time series classification approaches in response to these limitations. The potential of deep learning lies in its ability to capture complex patterns and hierarchies of attributes, and it has proven effective in addressing the challenges of time series classification, especially when dealing with datasets that contain a large number of training labels. For an in-depth exploration of time series classification using deep learning, we recommend the recent survey by *Herrmann & Webb (2021)*. An aspect of this field that adds an element of intrigue, and is not covered in the aforementioned (*Ali Nemer et al., 2022*), is the use of GNN for time series classification. By transforming time-series data into graph representations, the robust capabilities of GNNs can be used to capture patterns at both localized and overarching levels. Capturing intricate relationships between different samples of time series data within a given dataset is a strength of GNNs. The conversion of a single variable time series into a graph using the Series2Graph methodology aims to recognize unique patterns that enable precise classification using a GNN. In this method, each time series is considered as a graph, where the graph acts as input to the GNN and produces outputs for classification.

First, the decomposition of each series into subsequences produces nodes, and these nodes are connected by edges to represent their relationships. After this transformation, a GNN is used for graph classification. The Time2Graph+ approach first introduced the Series2Graph perspective (*Zhang et al., 2020b*). The Time2Graph + modeling procedure follows a two-step approach: first, a time series is transformed into a graph of shape lets, and then a GNN is used to understand the connections between these shape lets. The Time2Graph algorithm divides each time series into sequential segments when creating a shape-let graph. It then uses data mining techniques to assign characteristic shape lets to these subsequences. These shapes let you act as nodes within the graph, establishing connections through edges. The formation of edges between nodes is determined by the likelihood of a shape to occur sequentially in a given time series.

As a result, the transformation of each time series produces a graph in which the nodes are shape lets, and the edges are the transition probabilities between these shape lets. Following the construction of the graph, *Zerveas et al. (2021)* uses a graph attention network in conjunction with a graph pooling operation to extract the overall representation of the time series. The resulting representation is then fed into a classifier to assign class labels to the time series.

# ADVANTAGES AND LIMITATIONS OF GRAPH NEURAL NETWORKS

GNNs are greatly successful in the industrial sector due to their competence to model dense relational structures, compelling them standard for applications such as industrial IoT (IIoT), supply chain optimization, and predictive maintenance. Contrasting to traditional techniques, which strain interconnected data, GNNs certainly attain dependencies involving nodes, permitting more anomaly detection and precise predictions in industrial systems. For instance, in predictive maintenance, GNNs can investigate sensor data from interconnected technology to identify immediate signs of failure, curtailing expensive downtime. In supply chain management, they optimize logistics by modeling relationships between suppliers, warehouses, and distributors, reducing disruptions and developing efficiency. Furthermore, GNNs scale successfully to large networks, such as power grids, where they can predict failures and improve energy distribution. Although computationally demanding, they are substantially more competent than fully connected deep networks when dealing with structured data. Their capability to learn from graph-based dependencies, pooled with enhanced interpretability, makes GNNs a needed tool for adopting in industrial challenges that concern complex, interconnected systems.

*Li et al. (2018)* proposes a novel technique for modeling spatial dependencies in traffic flows. It fuses sequence-to-sequence framework, diffusion convolution, and scheduled sampling practice to capture both temporal and spatial dependency. However, the limitations consist of the lack of interpretability and complexity of deep neural networks, likely costly computation, and the dependence on a precise representation of the road network graph. *Gao & Ribeiro (2022)* introduces a time-then-graph architecture for time-based graphs and demonstrates a fluency advantage over time and graph. Although

better performance is achieved for certain tasks, limitations include an emphasis on certain events, possible inadequacies for edge-level prediction, and the need for further theoretical investigation. Research in *Pan et al. (2019)* presents a common house of attributed graphs, but expresses limitations in treating significant generalization as a result of lack of elementary operators of mathematics. The sequences generated by graphs have vital complexity and are constrained by hidden variables, posing challenges in grasping complicated dependencies. The study of *Zambon et al. (2019)* introduces a technique to predict traffic flow using the spatio-temporal graph CNN, a characteristic of this method is the adaptability to predict traffic flow. While successfully capturing active features, there are limitations when dealing with ordinary 2D or 3D network input and an explicit lack of attention to apparent inducing aspects such as weather. *Guo et al. (2019)* presents Graph WaveNet, an integration of dilated causal convolution with graph convolution for spatiotemporal graph simulation. Limitations arise from issues such as scalability and ambiguities in the training of hidden dependencies.

The research in *Wu et al. (2019)* focuses on traffic prediction using the Multi-Range Attentive Bicomponent, which successfully captures both the node and the edge relationships. Interpretability is challenged due to the complexity of the model and the study is deficient in showing training-time insights. In addition, *Chen et al. (2020)* proposes a model for traffic prediction that implements a structure learning convolution, involving convolutional neural networks in graph domains. Limitations arise in the areas of hyperparameter sensitivity and computational complexity. Moving forward, the research of *Wang et al. (2020b)* presents synchronized graph convolutional neural networks that can capture both spatial and temporal dependencies in different network data predictions. Despite consistently outperforming benchmarks, the challenges lie in performance differences due to dataset specificity and a lack of detail on internal mechanisms. The research proposed by *Zheng et al. (2020)*, data adaptive graph generation node adaptive and parameter learning modules for traffic prediction.

The technique achieves high-tech effectiveness, but challenges include a lack of thorough model evaluations and a comprehensive investigation of internal mechanisms. *Zhang et al. (2020a)* presents a mechanism to dwell on the issues of long-term graph convolutional networks and the differentiable architecture search for both spatial and temporal traffic forecasting. The study lacks a comprehensive analysis of the model, including the need to compare the model with existing state-of-the-art models. On the other hand, research by *Song et al. (2020)* proposes a spatio-temporal graph based on transformers for crowd trajectory prediction, relying exclusively on attention techniques. Although successful, limitations include potential degradation in predicting impulsive activity and challenges in handling constant time series sequential data. In summary, each study presents novel methods and frameworks for various time series prediction tasks, but these studies also express common limitations, such as problems of interpretability, model complexity, and limitations in dealing with external aspects or diverse datasets.

More comprehensive and comprehensive surveys are needed to improve the applicability and understanding of these techniques. The study in *Yu et al. (2020)* introduces a novel graph neural network with an output layer to focus on predicting graph

controls for graph-to-graph training. Graph controls or edits offer a formative and comprehensive illustration for innumerable changes in graph constructions, reviving computational competence and interpretability compared to prevailing approaches. Graph edits are well thought out and valuable in terms of sparsity. However, the method recognizes the limitation of the Markovian postulation, which implies imminent research to discover alternative frameworks. The need to fuse nonstructural data through alternative designs is also recognized. The scope of evaluation is inadequate and requires development in complex and diverse situations. The study by *Paassen et al. (2020)* presents graph convolutional networks with a time-aware zigzag topological layer for time-dependent graph neural networks.

The addition of zigzag trajectories increases the system's ability to predict time series on graphs, which means higher performance on different datasets. Nevertheless, its efficiency may be inadequate for time-dependent structures, and limitations related to mathematical context and generalizability are not systematically discovered. In *Chen, Segovia & Gel (2021)*, a framework for predicting traffic flow has been developed using a data-driven approach to build dynamic graphs with spatio-temporal dependencies. Although the techniques are successively applied, their generalizability to other domains or tasks is not categorically established. The reliance on past data to make the spatio-temporal distance hyperparameter sensitive is illustrative and heralds possible robustness concerns. Research conducted in *Lan et al. (2022)* introduced a graph module known as a temporal polynomial to capture the correlation representation of dynamic variables in the prediction of multivariate multidimensional time series. It supports limitations associated to real-world procedures, causal construction, and causal knowledge. The emphasis on adhering to these limitations is projected for future work.

Furthermore, in *Liu et al. (2022)* the authors introduce a hybrid model that integrates spatio-temporal graphs with neural controlled differential equations for traffic prediction. While superior in performance to reference models, it has a clear deficiency in detailing its comparative study and intuitions into the hyper-parameter sensitivity and the interface between neural-controlled differential equations and graph-controlled networks. *Choi et al. (2022)* presents a pre-training technique called TSFormer for long-term evidence learning in GNN. It successfully addresses the method's limitation in capturing long-term dependencies. However, scalability concerns remain and the characterization of "long-term" needs further investigation. A study by *Shao et al. (2022)* discusses the prediction of variable subsets and presents a technique that uses a non-parametric wrapper.

Although competent, its performance is inclined by hyperparameters, and the postulation of variables preoccupied at casual may not embrace continuously (*Jin et al., 2024*). Scalability constraints associated with large datasets are not widely discovered. Graph structures in a regularized framework by *Chauhan et al. (2022)*, integrate implicit and explicit graphical structures in a regularized framework to handle multivariate time series prediction. The technique's understanding of hyper-parameters and the configuration of cultivated graph structures with interpretability are accredited. While scalability remains unexplored. In the research conducted by *Yu et al. (2022b)*, they introduce a framework that learns temporal graphs from multivariate time series and that

too at manifold scales (*Chai et al., 2024*), overcoming static variable importance modeling constraints. However, it lacks further analysis of interpretability, scalability, and computational complexity properties (*Li et al., 2024*). The work of *Cui et al. (2021)* presents an innovative structure for detecting anomalies in multivariate time series using the graph attention network. In the run through, each research presents novel methods to detect anomalies in multivariate time series (*Qin et al., 2024*), addressing numerous features such as relationship learning (*Guo et al., 2024*), graph-based modeling, and anomaly metric strategy. Although all the work presented acknowledges positive limitations, additional analysis and detailed exploration is needed to improve the reasoning behind these innovative approaches and their applicability in the real world.

# ENHANCING MODEL INTERPRETABILITY AND EXPLAINABILITY

Interpretability and explainability are important to ensure understanding and trust in AI models (*Bell et al., 2022*), particularly in the framework of convoluted constructions such as DNNs and GNNs. In this segment, we explore approaches intended to enhance the interpretability and explainability of such models, in view of current progressions and dedicated tools custom-made for DNNs and GNNs.

## Model simplification and feature importance

An approach to improving interpretability is to simplify the model techniques. For DNNs, practices such as distillation, feature reduction, and pruning can support the generation of more interpretable techniques without significantly affecting performance (*Gosiewska, Kozak & Biecek, 2021*). Likewise, in GNNs, custom-made generalization plans for graph structures can be helpful in understanding the model's decision-making method. Furthermore, feature importance methods such as SHapley Additive Explanations (SHAP) and Local Interpretable Model-agnostic Explanations (LIME) can deliver understanding into the role of distinct features to the model's predictions, by this means setting off interpretability (*Gosiewska, Kozak & Biecek, 2021*).

## *Post-hoc* explanation methods

The objective of these methods is to provide interpretable intuitions into model predictions without altering the fundamental architecture. Methods such as occlusion analysis, saliency maps, and gradient-based techniques can provide significant explanations for DNNs by stressing vital sections of input data that affect output (*Han, Srinivas & Lakkaraju, 2022*). In the same way, in GNNs, techniques like graph visualization mechanisms and graph attention techniques can clarify the rank of nodes and edges in decision-making methods.

## Attention techniques and layer-wise relevance propagation

Attention techniques have appeared as commanding tools to set the stage for interpretability in DNN models. By robust weighting of input features built on their importance to assignment, attention techniques offer spontaneous enlightenment for model predictions (*Achtibat et al., 2024*). With reference to GNNs, attention techniques

can help categorize significant nodes and edges in the graph, enabling an improved understanding of the model's performance. Another technique, layer-wise relevance propagation, points relevance scores to separate layers of the neural network, subject to understanding of how input features assist to the output.

### Counterfactual explanations

These explanations provide an exclusive perspective on model interpretability by supplying explanations for distinct predictions planted on hypothetical states. By making counterfactual instances that advance to diverse results, these practices help users recognize the decision limitations of the model and categorize actionable understandings to develop fairness and model performance (*Guidotti, 2022*).

### Tools for model interpretability

Quite a few dedicated libraries and tools have been established to enable model interpretability for DNNs and GNNs. For example, libraries including Captum and TensorFlow Explainability offer the use of many customized explanation methods for DNN models (*Kaur et al., 2020*). Similarly, tools such as the Deep Graph Library provide services for interpreting and visualizing GNNs, making them more accessible to practitioners and researchers. Integrating these tools and approaches into the evaluation and development of DNNs and GNNs can meaningfully enhance their explainability, interpretability, nurturing trust, and adoption in applications associated with the real world (*Bell et al., 2022*). As the field progresses, more research into customized innovative frameworks and techniques for multipart models will be important to endorse accountability and transparency in AI systems.

## ETHICAL CONSIDERATIONS, SECURITY AND DATA PRIVACY CHALLENGES

Although improving model interpretability and scalability is decisive, it is also imperative to consider ethical issues, security challenges, and data privacy, particularly in the context of time series analysis in industrial frameworks (*Albahri et al., 2023*). This section dives into the ethical inferences of implementing AI practices, the prerequisite for preserving data privacy, and recommends practices and best strategies to direct these compound issues reliably. Implementing AI techniques in industrial contexts can have reflective ethical consequences, mainly when decisions based on these techniques influence entities or communities (*de Almeida, dos Santos & Farias, 2021*). Biases in the unintended consequences, algorithmic fairness, and training data of model predictions are some serious ethical issues that require careful consideration (*Beil et al., 2019*). It is important to certify that models are accountable, fair and transparent to moderate latent harms and safeguard impartial consequences for all participants. Modern AI techniques, exclusively DNN architectures, regularly need immense data for training, raising concerns about data privacy and security (*Zareen, Akram & Ahmad Khan, 2020*). Time series in industrial frameworks can contain sensitive data, such as confidential business, personal identification, or actions of proprietary processes (*Díaz-Rodríguez et al., 2023*). Protecting

these data against breaches, misuse, or unlicensed access is paramount in ensuring compliance and trust with control frameworks such as the General Data Protection Regulation and the California Consumer Privacy Act (*Alexander, 2019*).

Best practices and guidelines that integrate data privacy, security, and ethical considerations in the deployment and development of AI models require a positive adherence and method to the best practices. For example: Documentation of data sources, model architecture, and decision-making processes needed to be present to support accountability and transparency. Reduce the retention and collection of personally identifiable information to minimize the risk of unauthorized access and privacy breaches (*Goldsteen et al., 2022*). Consider methods such as differential privacy and data anonymization to protect sensitive data while preserving the utility of the information for model training and analysis (*González-Sendino, Serrano & Bajo, 2024*). Ensure that these AI models are free from interpretable and explainable biases to encourage fairness and reduce the risk of accidental consequences (*Omotunde & Ahmed, 2023*). Implement robust security actions, including encryption, auditing, and access controls, to protect data against unauthorized access and cyber threats (*Borgman, 2018*). Acquire up-to-date consent from users concerning sharing practices, data collection, and usage, also make available transparency about how their data is being used to deploy and build AI models.

## OPEN QUESTIONS, CHALLENGES, AND PROPOSED SOLUTIONS

From our investigation, it is apparent that although the progress in time-series research is still quite extensive, there are still loads of open issues and research questions to be addressed. It is evident that further studies are needed in most of the industrial approaches in the context of time series. Hence, adding together to the already research work, the following are still complementary to open gaps in research.

### High-dimensional anomaly detection

Identifying anomalies in high dimensional data is challenging because of not identifying precise local correlations, thus using techniques such as UMAP, t-SNE, or autoencoders, can be employed to convert high-dimensional data into lower-dimensional illustrations, preserving important correlations whereas dropping computational complexity. In addition, algorithms like graph-based anomaly detection that take advantage from the correlations between dimensions can be investigated. Also, incorporating attention mechanisms with current anomaly detection techniques can vigorously spotlight significant features, focusing on the challenge of unsuitable feature selection.

### Ensemble methods

Traditional ML techniques such as clustering usually fail to provide robust outcomes for anomaly detection. Ensemble techniques, which merge outputs from multiple methods, deliver a favorable alternative. Hence, finding methods to choose the base learners and accurate subspaces is valuable. In addition, deciding on the right combination and quantity policies is still a challenging and open research problem to adopt. Implementing

heterogeneous ensemble learning a combination of diverse methods (*e.g.*, SVMs, decision trees, and neural networks) to leverage their original strengths. Stacking techniques or voting mechanisms can additionally improve anomaly detection accuracy. By developing adaptive ensemble techniques that dynamically choose subspaces and base learners constructed on data characteristics, enhancing both detection quality and runtime.

### Evolving data dimensions

The loss or arrival of existing or new dimensions during the period is yet another stimulating open research area for the future. In prospective application areas such as anomaly detection in IoT or industrial IoT devices, sensors can be on or off periodically, and innovative techniques are required to detect anomalies or outliers more competently in this challenging and stimulating scenario. By developing incremental learning algorithms that can easily adapt to modifications in data structure over different time periods. Methods such as adaptive LSTMs and online GNNs can deal with dynamic time-series data and evolving graph structure. Furthermore, incorporating techniques such as transfer learning can be adapted so that models trained on one set of dimensions to new data, reducing retraining struggles.

### Scarcity of public real-world datasets

Unlike data set sources for conventional ML problems for time series, publicly available real-world data sets for industrial data are very scarce. One key reason is that, in industry, data is believed to be extremely confidential—the data hold wear and tear or faults in their products reasonably for predictive maintenance. This has serious significance in the research work in this domain, because of public non availability of real-world data, research is classically steered by academia in association with industry, manufacturers, or suppliers, and in companies themselves. Hence, a non-efficiency in follow-up research and lack of reproducible research. Encouraging industry-academia cooperation to initiate anonymized datasets however making certain and preserving data privacy. Also, employing data synthesis approaches, such as generative adversarial networks (GANs), to make realistic industrial data for research. Exploring domain adaptation techniques that can transfer knowledge from accessible datasets to unseen, new industrial domains, minimizing reliance on explicit datasets.

### Lack of labeled data

There is a lack of labeled data in industry, operating with datasets of real-world industry propose the prospect of being capable of assessing developed techniques in practice. Consequently, real-world data is frequently not or only incompletely labelled, since to annotate this volume of data requires expert knowledge and is hugely time-consuming. However, employing self-supervised learning and semi-supervised learning approaches to take advantage of unlabeled data. As well, techniques like pseudo-labeling and contrastive learning can improve model performance without the need of extensive labeling. In

addition, developing annotation bases employing active learning, where models rank first the most useful data points for skilled labeling, optimizing annotation attempts.

### Representation of training data

Finding representative training datasets is quite challenging due to the scale and variety of industrial applications and its products. Using simulated data from software-in-the-loop systems to accompany real-world data. Exploring domain-specific data augmentation and GANs approaches to improve dataset representativeness and diversity. Designing algorithms that are competent of leveraging synthetic data while easing domain differences using methods like domain adversarial neural networks (DANNs).

### Model interpretability

The issue with non-interpretability of complex ML models demands details for the replacement of components. With maintenance that is interval-based the justification is insignificant; conversely, with utilizing advanced ML techniques, *e.g.*, deep learning models, interpretation and explanation of the techniques and their choices turn into a challenge. By utilizing Explainable AI (XAI) methods such as LIME, SHAP, and attention visualization to present understandings into model decisions. Aim on acquiring interpretable designs like prototype-based learning models. Coming to GNNs, it is quite challenging to interpret them critical industrial systems because of opaque decision process, regulatory compliance, and trust in automation. For instance, to trace certain decisions input will be quite difficult in GNNs, in healthcare or energy industry they demand XAI for accountability, and industrial stakeholders require transparency to trust GNNs. However, we can use attention mechanisms for influential nodes and edges in GNNs, also the use of XAI techniques as discussed earlier is advisable, and employ graph visualization tools. Furthermore, exploring hybrid techniques which combines interpretable ML approaches with deep learning to balance explainability and accuracy. For example, incorporate rule-based systems or decision trees with neural networks to increase interpretability.

By focusing on these challenges with far-forward techniques and algorithms, the research society can substantially augment the scalability, robustness, and applicability of time-series in industrial frameworks.

## CONCLUSION

In summary, this comprehensive survey acts as a guide to the vast field of time-series analysis within industrial data. It starts with the basic principles of traditional machine learning, which provided insights for forecasting, demand management, and production optimization. As the industrial landscape evolved, the integration of deep learning architectures led to a transformative phase. ML, DL, and GNN approaches have shown great promise in various fields, as well as in time series. To run through the past, investigate the present, and analyze the future, in this survey, we made several contributions to the application of ML, DL, and GNNs theory in the domain of time series. The merits and

limitations of DL and GNN were evaluated in detail compared to traditional ML approaches. In addition, the tendencies of industrial frameworks were considered in detail to explain the significance and necessity of AI algorithms for time-series modeling. Key ML, DL, and GNN models and frameworks were summarized and reviewed to facilitate the use and development of AI-based tools. Despite the contributions of ML, DL, and GNNs limitations, exits that open doors for further investigation. With DL and GNNs interpretability remains the issue, their "black-box" nature remains a significant challenge, methods should focus on integrating XAI techniques. GNNs hold promise, but it has issues in adaptability and scalability in various industrial scenarios, the focus should be shifted to domain specific adaptations. The scarcity of labeled, publicly available and representative industrial data hinders benchmarking and reproducibility; addressing this problem requires collaborative efforts between industry and academia. Promising research questions and points for future run were explored shortly. However, challenges remain, including concerns about the interpretability of deep learning models and the need for graph-based solutions tailored to different industrial applications.

## ACKNOWLEDGEMENTS

We have used ChatGPT for sentence structuring and any grammatical mistakes.

### Funding

The work received funding from the European Union Horizon 2020 research innovation, the SDGine program under the Marie Skodowska-Curie grant agreement number 945139. This work was also funded by the European Commission through the HORIZON-JU-SNS-2022 ACROSS project with Grant Agreement number 101097122. There was no additional external funding received for this study. The funders had no role in study design, data collection and analysis, decision to publish, or preparation of the manuscript.

### Grant Disclosures

The following grant information was disclosed by the authors:
European Union Horizon 2020 Research Innovation.
SDGine Program under the Marie Skodowska-Curie: 945139.
European Commission through the HORIZON-JU-SNS-2022 ACROSS: 101097122.

### Competing Interests

The authors declare that they have no competing interests.

### Author Contributions

- Muhammad Jamal Ahmed conceived and designed the experiments, performed the experiments, analyzed the data, performed the computation work, prepared figures and/or tables, authored or reviewed drafts of the article, and approved the final draft.

- Alberto Mozo conceived and designed the experiments, analyzed the data, performed the computation work, authored or reviewed drafts of the article, and approved the final draft.
- Amit Karamchandani analyzed the data, performed the computation work, authored or reviewed drafts of the article, and approved the final draft.

## Data Availability

This is a literature review.

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
