# Peer review of "A survey on graph neural networks, machine learning and deep learning techniques for time series applications in industry"

_PeerJ Computer Science, doi:10.7717/peerj-cs.3097_

## Round 0.1 · original submission · Major Revisions

· Academic Editor

Major Revisions

You have received three reviews for your paper. The reviews are unfortunately mixed, with two reviewers suggesting Major revision and one reviewer (R1) suggesting rejection. The reviewers raise multiple issues regarding the introduction (motivation and justification of choices), literature review (research design) and the reporting of the results. I would also like to point out that the structure of the paper can be improved, it is not very clear why the subsections are required per technology. Also, I miss a detailed comparison of the technologies (performance of different datasets, run time etc.).

Please address all the review comments, and upload a detailed reply as part of your revision.

I wish the authors all the very best for their revision.

Reviewer 1 ·

Basic reporting

Your paper addresses an important and timely topic, investigating Artificial Intelligence (AI) in the context of time series data applied to the industrial domain. It explores various dimensions, including classical Machine Learning (ML), Deep Neural Networks (DNNs), and Graph Neural Networks (GNNs). While the paper covers a relevant subject within the scope of this journal, several areas require improvement to enhance its clarity, focus, and methodological rigor.

Overall Feedback:

+ While the manuscript is generally well-structured, the writing could benefit from more careful proofreading to address grammatical and syntactical issues. A thorough revision will significantly improve the readability and impact of your work.

+ I recommend providing an external repository containing the BibTeX and metadata used in your analysis to ensure transparency and reproducibility.

+ Some figures are included in the manuscript but are not referenced in the text. All figures should be appropriately cited and discussed within the corresponding sections.

Experimental design

Major Concerns:

1. Lack of Scope and Justification:

The study lacks focus and a clear justification for its choices. For example, the discussion around time series data remains too broad, without specifying particular applications such as anomaly detection, classification, or distance measures. Additionally, the use of the term "industry" is vague. Narrowing the scope to a more specific area, such as fault detection or diagnostics, could strengthen the paper’s focus. Furthermore, it would be useful to clarify whether the paper relates to Industry 4.0, a widely discussed concept in current literature.

2. Selection of Dimensions:

The paper surveys three dimensions: classical ML, DNNs, and GNNs. However, the rationale for selecting these specific dimensions is unclear. For instance, GNNs can be seen as a subfield of machine learning, so it is unclear why they are treated separately. The distinction between these categories, particularly the division of deep learning, is not well justified. I encourage the authors to reconsider this structure or provide a more robust explanation for these choices.

3. Methodological Rigor:

The methodology section is underdeveloped and must be thoroughly rewritten. Key details are missing, such as the number of articles reviewed, the inclusion and exclusion criteria, the time range of the literature search, and the databases chosen. The current methodology does not meet the standards of a systematic review. Furthermore, including repositories like Google Scholar and ArXiv without explaining the rationale behind these choices undermines the rigor of the methodology. I recommend specifying formal selection criteria, particularly for unsupervised models and semi-supervised learning (SSL), to enhance transparency.

4. Related studies

It is necessary to cite related studies or reviews that are close to the proposal of this article and what the gaps in these studies are.

Validity of the findings

Detailed Section Feedback:

1 .Introduction:

The introduction is too long and reads more like a comprehensive summary of the entire paper. A more concise introduction should present the research problem, the gap in the literature, a brief outline of the methodology, and a summary of key findings. Detailed discussions belong in the following sections. Moreover, the introduction transitions between topics in a somewhat disjointed manner, making the timeline of ideas difficult to follow.

Specific points:

+ Line 76: The authors discuss the contribution of the paper abruptly in the middle of a paragraph, which causes confusion. This should be introduced more clearly.
+ Line 86: The phrase "Apparently signified by ARIMA" lacks clarity. Why is ARIMA only "apparently" significant?
+ Line 92: The phrase "Recent advances in conventional machine learning" is followed by a citation from 1997, which contradicts the notion of "recent" advances.
+ Line 142: The text seems to shift topics abruptly, disrupting the flow of the discussion.

2. Methodology:

As mentioned earlier, this section lacks the necessary detail for reproducibility. Please include:

+ The number of articles collected and those rejected, along with the reasons for rejection.
+ The time range of the literature reviewed.
+ The generic search strings used (instead of examples).
+ The specific selection criteria for unsupervised models or SSL approaches. Additionally, the rationale for selecting certain databases (e.g., why Google Scholar?) and the exclusion of others (such as ArXiv) should be clarified. As it stands, the current methodology does not meet the requirements for a systematic review.

3. Applications of Time Series Analysis:

The manuscript would benefit from a section outlining the basic concepts of time series analysis and how they relate to machine learning. This would help set the foundation for the more advanced topics discussed later in the paper.

4. Open Questions:
The manuscript does not clearly list the open issues in the field or provide guidelines for future research directions. This section should be expanded to highlight unresolved challenges and suggest possible areas for further exploration.

5. Conclusion:
The conclusion requires further development. Specifically, it should address the limitations of the proposed method and revisit the contributions mentioned in the introduction, summarizing them in light of the findings. This will help to reinforce the paper’s main contributions and discoveries.

Additional comments

Your paper addresses an important and timely topic, investigating Artificial Intelligence (AI) in the context of time series data applied to the industrial domain. It explores various dimensions, including classical Machine Learning (ML), Deep Neural Networks (DNNs), and Graph Neural Networks (GNNs). While the paper covers a relevant subject within the scope of this journal, several areas require improvement to enhance its clarity, focus, and methodological rigor.

Overall Feedback:

+ While the manuscript is generally well-structured, the writing could benefit from more careful proofreading to address grammatical and syntactical issues. A thorough revision will significantly improve the readability and impact of your work.

+ I recommend providing an external repository containing the BibTeX and metadata used in your analysis to ensure transparency and reproducibility.

+ Some figures are included in the manuscript but are not referenced in the text. All figures should be appropriately cited and discussed within the corresponding sections.

Reviewer 2 ·

Basic reporting

Extensive studies have been conducted to investigate Artificial Intelligence (AI) in the context of time series data. In this article, we investigate the complex domain of industrial time series, from the dimensions of classical Machine Learning (ML), Deep Neural Networks (DNNs) and Graph Neural Networks (GNNs). Current surveys often focus on a specific methodology or oversee the connection of diverse approaches; our article bridges this gap by providing an all-inclusive interpretation across numerous techniques. In addition, this article aims to focus on the core areas of time series such as forecasting, classification, and anomaly detection. From traditional methodologies like Autoregressive Integrated Moving Average (ARIMA) and Support Vector Machine (SVM) methods, the advancements of DNNs, for instance Long-Short-Term Memory (LSTMs), Convolutional Neural Networks (CNNs), attention mechanisms, and transformers, describe how temporal information is used for forecasting, anomaly detection, and classification. Then the article discusses the advances and limitations in ML, DNN, and GNN in order to improve the different methods in either category. Lastly, we outline future directions and open research questions with the different methodologies used in time series.

Experimental design

The article lacks some of the requirements related to publishing in this journal.
Comments
1. There is no discussion of datasets that are used in the approaches regarding Machine Learning(ML), Deep Neural Networks(DNNs) and Graph Neural Networks (GNNs).
2. There must be a comparison of approaches of ML, DNNs and GNNs in tabular form that depicts clearly the core areas of time series such as forecasting, classification and anomaly detection.
3. The paper needs to be proofread, also correction for the structure of the sentence is required.
4. Figure 01 needs to be redrawn to clearly indicate the applications of big data in the industrial sector.

Validity of the findings

It lacks comparison of different studies and discussion about the limitations

Additional comments

Extensive studies have been conducted to investigate Artificial Intelligence (AI) in the context of time series data. In this article, we investigate the complex domain of industrial time series, from the dimensions of classical Machine Learning (ML), Deep Neural Networks (DNNs) and Graph Neural Networks (GNNs). Current surveys often focus on a specific methodology or oversee the connection of diverse approaches; our article bridges this gap by providing an all-inclusive interpretation across numerous techniques. In addition, this article aims to focus on the core areas of time series such as forecasting, classification, and anomaly detection. From traditional methodologies like Autoregressive Integrated Moving Average (ARIMA) and Support Vector Machine (SVM) methods, the advancements of DNNs, for instance Long-Short-Term Memory (LSTMs), Convolutional Neural Networks (CNNs), attention mechanisms, and transformers, describe how temporal information is used for forecasting, anomaly detection, and classification. Then the article discusses the advances and limitations in ML, DNN, and GNN in order to improve the different methods in either category. Lastly, we outline future directions and open research questions with the different methodologies used in time series.
Comments

1. There is no discussion of datasets that are used in the approaches regarding Machine Learning(ML), Deep Neural Networks(DNNs) and Graph Neural Networks (GNNs).
2. There must be a comparison of approaches of ML, DNNs and GNNs in tabular form that depicts clearly the core areas of time series such as forecasting, classification and anomaly detection.
3. The paper needs to be proofread, also correction for the structure of the sentence is required.
4. Figure 01 needs to be redrawn to clearly indicate the applications of big data in the industrial sector.

5. The literature needs to be updated by adding the latest research papers of the year 2024. i.e.,

• Jin, W., Tian, X., Shi, B., Zhao, B., Duan, H.,... Wu, H. (2024). Enhanced UAV Pursuit-Evasion Using Boids Modelling: A Synergistic Integration of Bird Swarm Intelligence and DRL. Computers, Materials & Continua, 80(3), 3523-3553. doi: https://doi.org/10.32604/cmc.2024.055125
• Chai, S., Wang, S., Liu, C., Liu, X., Liu, T.,... Yang, R. (2024). A visual measurement algorithm for vibration displacement of rotating body using semantic segmentation network. Expert Systems with Applications, 237, 121306. doi: https://doi.org/10.1016/j.eswa.2023.121306
• Ren, W., Jin, N., & OuYang, L. (2024). Phase Space Graph Convolutional Network for Chaotic Time Series Learning. IEEE Transactions on Industrial Informatics, 20(5), 7576-7584. doi: 10.1109/TII.2024.3363089
• Li, T., Hui, S., Zhang, S., Wang, H., Zhang, Y., Hui, P.,... Li, Y. (2024). Mobile User Traffic Generation Via Multi-Scale Hierarchical GAN. ACM Trans. Knowl. Discov. Data, 18(8), 1-19. doi: https://doi.org/10.1145/3664655
• Qin, C., Shi, G., Tao, J., Yu, H., Jin, Y., Xiao, D.,... Liu, C. (2024). RCLSTMNet: A Residual-convolutional-LSTM Neural Network for Forecasting Cutterhead Torque in Shield Machine. International Journal of Control, Automation and Systems, 22(2), 705-721. doi: https://doi.org/10.1007/s12555-022-0104-x
• Guo, T., Yuan, H., Hamzaoui, R., Wang, X., & Wang, L. (2024). Dependence-Based Coarse-to-Fine Approach for Reducing Distortion Accumulation in G-PCC Attribute Compression. IEEE Transactions on Industrial Informatics. doi: 10.1109/TII.2024.3403262
• Liu, K., Jiao, S., Nie, G., Ma, H., Gao, B., Sun, C.,... Wu, G. (2024). On image transformation for partial discharge source identification in vehicle cable terminals of high-speed trains. High Voltage, 1-11. doi: https://doi.org/10.1049/hve2.12487

6. The advantages and limitations of Machine Learning, Deep Neural Networks and Graph Neural Networks are not clear, the terms need more explanation in order to make it more informative.
7. The paper needs to be rechecked in order to match the approaches of ML, DNNs and GNNs used for industrial time series or cover the other aspects.
8. There is no explanation nor comparison of these approaches ML, DNNs and GNNs that depicts which of these are better used to find the time series application in the industrial sector.
9. In the future directions the problems or open research issues are mentioned but there is no discussion of solutions to these problems with improved algorithms.

Annotated reviews are not available for download in order to protect the identity of reviewers who chose to remain anonymous.

·

Basic reporting

The manuscript presents a comprehensive review of three families of models—Machine Learning (ML), Deep Learning (DL), and Graph Neural Networks (GNNs)—and their applications to industrial time series tasks such as forecasting, classification, and anomaly detection. The topic is timely, especially with the rise of Industry 4.0 applications and interconnected systems. However, while the paper covers these models, the distribution of content and focus on GNNs relative to the other two types of models could be more balanced to align with the title and core objectives. The manuscript claims to focus equally on ML, DL, and GNN techniques for time series applications, but GNN-related content seems somewhat underdeveloped compared to the detailed discussion on LSTMs, CNNs, and traditional ML models. Additionally, include more industrial case studies that highlight the unique advantages of GNNs in handling interconnected processes or graph-structured time series data..

Experimental design

Observation:
While the manuscript touches on various models (ARIMA, LSTM, CNN, and GNN), it could clarify the specific roles each type plays in industrial applications.
Suggestion:
Include a comparison table or diagram to illustrate:
Where traditional ML (e.g., ARIMA) is preferred (e.g., for interpretable forecasting).
When DL models (like LSTMs or CNNs) perform better (e.g., for non-linear dependencies).
Why GNNs are more suitable for handling interconnected data in real-world industrial systems.
While the survey mentions both traditional NNs and GNNs, it lacks a comparative analysis highlighting the unique strengths of GNNs in industrial settings. Readers will benefit from a clearer understanding of when and why GNNs are preferable over LSTMs or CNNs.
Add explicit research questions and use PRISMA for quality assessment.

Validity of the findings

Identify Gaps and Future Research Directions More Clearly
Observation:
The paper touches on some open challenges but does not discuss gaps for time series applications.
Suggestion:
Expand the future directions section to discuss:
The need for scalable GNNs in industrial environments.
Challenges of interpreting GNNs in critical industrial systems.
The potential of hybrid GNN-DL models for time series forecasting

---

## Round 0.2 · Major Revisions

· Academic Editor

Major Revisions

The reviewers have commented on your paper, and two reviewers still require Major revisions.

Please note that all reviewers need to accept your paper for the final acceptance. Please attach a detailed document with your responses that explains all the edits and changes done to your article.

Reviewer 1 ·

Basic reporting

The authors have improved the text; however, the methodology remains unreliable and lacks reproducibility.

It is essential that the methodology section clearly describes the entire process undertaken in the systematic review to identify applications and dimensions. For instance, how did the authors identify the papers related to a specific application? If the review is "systematic", there must be a categorization process, allowing for a count of the papers associated with each category. A systematic approach implies a well-defined procedure, which should be explicitly outlined, step by step, for each discovery.

The methodology section is underdeveloped and requires substantial revision. Several critical details are missing, such as the total number of articles reviewed, the period of the literature search, and other key methodological aspects. In its current state, the methodology does not meet the standards of a systematic review.

This section lacks the necessary level of detail to ensure reproducibility. To address this issue, the authors should include a table with the following information:

+ The number of articles collected, those excluded, and the corresponding reasons for exclusion.
+ The period covered in the literature review as inclusion or exclusion criteria.
+ The generic search strings used for each database, rather than only providing examples.
+ A link to an external repository containing the BibTeX and metadata used in the analysis to enhance transparency and reproducibility.
+ A graph showing the distribution of studies by year.
+ The number of studies categorized by application and dimension.
+ Other important information in a systematic review.

Experimental design

no comment

Validity of the findings

no comment

Additional comments

no comment

Reviewer 2 ·

Basic reporting

The revised version is improved.

Experimental design

The authors have improved the system design according to comments.

Validity of the findings

They have included comparison of different studeis.

Additional comments

The revised version of the paper may be accepted for publication.

·

Basic reporting

no comment

Experimental design

1. Include a PRISMA diagram to illustrate the study selection process, including the number of studies identified, screened, and included in the final analysis.
2. Clearly define inclusion and exclusion criteria for the studies selected for this survey.

Validity of the findings

3. Balance theoretical and practical content by incorporating more real-world examples and data to support your arguments.
4. Enhance table quality and citation by ensuring that tables are informative, well-designed, and properly cited, with clear references for all data presented.
5. Innovate and contribute technically by exploring new ideas, methodologies, or applications that can advance the field, providing novel insights and justifications for key claims.
6. Remove repetitive content, particularly sections related to interpretability and scalability, to streamline your narrative.
7. Employ performance metrics for comparison to provide a clear understanding of the strengths and weaknesses of different approaches.
8. Justify key claims, such as the superiority of Graph Neural Networks (GNNs), with robust arguments that are well-supported and convincing.

Additional comments

no comments

---

## Round 0.3 · accepted · Accept

· Academic Editor

Accept

Congratulations, your paper has been accepted for publication.

Reviewer 1 ·

Basic reporting

The authors have addressed all my comments, and I have no further suggestions.

Experimental design

no comment

Validity of the findings

no comment

Reviewer 2 ·

Basic reporting

The improved version is OK.

Experimental design

The improved version is OK.

Validity of the findings

The improved version is OK.